# FedWMSAM: Fast and Flat Federated Learning via Weighted Momentum and Sharpness-Aware Minimization

**Tianle Li**[*]
College of Computer Science and
Software Engineering
Shenzhen University
2200271015@email.szu.edu.cn

**Yongzhi Huang**[*]
Data Science Analysis
The Hong Kong University of Science
and Technology (Guangzhou)
huangyongzhi@email.szu.edu.cn

**Linshan Jiang**[†]
National University of
Singapore
linshan@nus.edu.sg

**Chang Liu**
Nanyang Technological
University, Singapore
liuc0063@e.ntu.edu.sg

**Qipeng Xie**
The Hong Kong University of Science
and Technology (Guangzhou)
qxieaf@connect.ust.hk

**Wenfeng Du**
Shenzhen University
duwf@szu.edu.cn

**Lu Wang**[‡]
Shenzhen University
wanglu@szu.edu.cn

**Kaishun Wu**[†]
The Hong Kong University of Science
and Technology (Guangzhou)
wuks@hkust-gz.edu.cn

## Abstract

In federated learning (FL), models must *converge quickly* under tight communication budgets while *generalizing* across non-IID client distributions. These twin requirements have naturally led to two widely used techniques: client/server *momentum* to accelerate progress, and *sharpness-aware minimization* (SAM) to prefer flat solutions. However, simply combining momentum and SAM leaves two structural issues unresolved in non-IID FL. We identify and formalize two failure modes: *local–global curvature misalignment* (local SAM directions need not reflect the global loss geometry) and *momentum-echo oscillation* (late-stage instability caused by accumulated momentum). To our knowledge, these failure modes have not been jointly articulated and addressed in the FL literature.

We propose **FedWMSAM** to address both failure modes. First, we construct a momentum-guided global perturbation from server-aggregated momentum to align clients' SAM directions with the global descent geometry, enabling a *single-backprop* SAM approximation that preserves efficiency. Second, we couple momentum and SAM via a cosine-similarity adaptive rule, yielding an early-momentum, late-SAM two-phase training schedule. We provide a non-IID convergence bound that *explicitly models the perturbation-induced variance $\sigma_\rho^2 = \sigma^2 + (L\rho)^2$* and its dependence on $(S, K, R, N)$ on the theory side. We conduct extensive experiments on multiple datasets and model architectures, and the results validate the effectiveness, adaptability, and robustness of our method, demonstrating its superiority in addressing the optimization challenges of Federated Learning. Our code is available at https://github.com/Li-Tian-Le/NeurlPS_FedWMSAM.

[*]Equal contribution.
[†]Co-corresponding author.
[‡]Main corresponding author.

39th Conference on Neural Information Processing Systems (NeurIPS 2025).

# 1 Introduction

Federated Learning (FL) [1] has emerged as a promising distributed learning paradigm that enables multiple clients to collaboratively train a shared global model while keeping their local data decentralized, thus preserving privacy. In particular, under the edge computing setting, FL has shown great potential in a wide range of real-world applications, including personal mobile sensing systems [2], healthcare data analytics [3, 4], and industrial Internet of Things scenarios [5–7].

FL in edge computing has several unique attributes. To reduce communication costs, the number of local iterations must increase. Concurrently, partial participation is often used due to the unstable connectivity of IoT devices. These two factors exacerbate the harmful effects of data heterogeneity [8–10] in real-world applications, intensifying client drift [11], where local updates deviate significantly from the global objective, degrading model performance and generalization.

Local multi-round training increases the drift caused by data heterogeneity, while partial participation further magnifies its effects. Based on where the drift originates, current methods for mitigating client drift can be categorized into the following levels: Methods at the data level attempt to balance data distribution through techniques such as data augmentation [12, 13] or resampling [8, 14], but these methods may incur high computational costs and risk overfitting. Gradient-level approaches, including proximal methods [9], gradient correction [11], and regularization terms [15], aim to adjust local gradients, but gradient distortion may hinder convergence. Techniques for model aggregation [16–18] alter the server's aggregation strategy, requiring the collection of additional information that may conflict with the privacy principles central to FL. At the same time, cryptographic PPFL (e.g., HE) offers an alternative at the cost of efficiency [19, 20].

Moreover, on the one hand, these methods focus solely on addressing client drift, with little consideration for convergence speed, while fewer training rounds could enhance real-world applicability. On the other hand, when heterogeneity is high, the loss surface resulting from aggregating models trained with Empirical Risk Minimization (ERM) becomes sharp, limiting the models' generalization in practical applications. Moreover, in non-IID FL, we observe two structural failure modes when simply porting SAM and/or momentum: (i) *local–global curvature misalignment*—local SAM directions need not reflect the global loss geometry; (ii) *momentum-echo oscillation*—late-stage instability caused by accumulated momentum. These motivate a design that can both *align local updates to the global geometry* and *adaptively damp momentum* as training progresses.

To solve this problem, we aim to design a *fast and flat* FL algorithm. Two mainstream lines address the twin goals in FL: *momentum* [21–25] for speed and *SAM* [26–28] for flatness. However, under non-IID data, each line has structural drawbacks: momentum can amplify late-stage instability/overfitting [29], while SAM *requires an extra backward pass* and, more importantly, its *local* perturbation directions need not reflect the *global* loss geometry (the local–global curvature misalignment diagnosed above). Momentum-only designs tailored to long-tailed heterogeneity (e.g., FedWCM [24, 25]) improve robustness but do not enforce global flatness.

A straightforward combination exists—MoFedSAM inserts SAM into FedCM [26, 22]—yet *naively* plugging SAM into a momentum pipeline leaves the two failure modes unresolved in non-IID settings (misalignment persists, late-stage momentum oscillation is not damped). This motivates a design that both *aligns local updates to the global geometry* and *adaptively damps momentum* as training progresses, we instantiate this next as **FedWMSAM**.

To address the aforementioned issues and achieve *fast and flat* training in FL, we propose **FedWM-SAM** (Federated Learning with Weighted Momentum–SAM), which integrates momentum with SAM in a principled way.

- **Firstly**, we introduce personalized momentum and use the *server-aggregated momentum* as a global geometric carrier to build a *momentum-guided global perturbation*, aligning local SAM directions with the global descent geometry; the perturbation is implemented with a *single backpropagation* (no extra backward pass).

- **Secondly**, we dynamically adjust the perturbation along this global direction during local steps (e.g., $\hat{x}_b^r = x_r + b\,\Delta_r$), enabling each client to explore globally flatter regions without increasing per-round cost.

- **Thirdly**, we *couple momentum and SAM* via a *cosine-similarity adaptive weight* $\alpha_r$, which yields an *early-momentum / late-SAM* two-phase schedule—speeding up early progress while damping late-stage momentum oscillations under non-IID data.

Our main contributions are summarized as follows:

- **Mechanism.** We identify and formalize two failure modes in non-IID FL—*local–global curvature misalignment* and *momentum-echo oscillation*—and correct them via a *momentum-guided global perturbation* and a *cosine-adaptive coupling*.
- **Method & Efficiency.** FedWMSAM aligns local SAM directions using *server-aggregated momentum* and implements the perturbation with a *single backpropagation*, yielding an *early-momentum / late-SAM* two-phase schedule and *fast-then-flat* training with near-FedAvg per-round cost.
- **Theory.** We provide a non-IID convergence bound that *explicitly models the perturbation-induced variance* $\sigma_\rho^2 = \sigma^2 + (L\rho)^2$ and its dependence on $(S, K, R, N)$: $\tilde{\mathcal{O}}\Big( \sqrt{\frac{L\Delta\,\sigma_\rho^2}{SKR}} + \frac{L\Delta}{R}\big(1 + \frac{N^{2/3}}{S}\big)\Big)$.
- **Empirics.** Across three datasets and twelve heterogeneity settings, FedWMSAM is best or on par in most cases and shows the largest gains under strong non-IID settings, reaching target accuracies in fewer rounds at similar per-round time, and the trends match our mechanism and theory.

## 2 Related Work

### 2.1 Heterogeneous Federated Learning

Federated learning (FL) often encounters the challenge of *client drift*. Existing solutions can be divided into three levels: data, gradient, and aggregation. At the data level, resampling strategies [14] adjust local sampling probabilities to balance class distributions, generative models like GANs [12] and VAEs [13] synthesize balanced datasets, and data-sharing approaches [8] distribute small portions of global data to clients. At the gradient level, methods such as SCAFFOLD [11] reduce variance using control variates, FedDyn [15] applies dynamic regularization, FedProx [9] stabilizes updates with proximal terms, and FedCM [22] aligns updates via momentum correction. At the aggregation level, Hierarchical FL [17] applies multi-level aggregation for large-scale networks, Clustered FL [16] groups clients with similar data distributions for localized optimization, and works like Client Selection [30] and Client Weighting [18] adjust the influence of clients based on their contribution to the global model. However, these works focus solely on addressing heterogeneity data, neglecting speed and model generalization.

### 2.2 Momentum-Based Federated Learning

Utilizing historical gradient information via momentum has proven effective for accelerating convergence and handling data heterogeneity in FL. The fundamental idea is to constrain current updates by past directions, smoothing out oscillations. MIME [21] and FedCM [22] compute a global momentum on the server and distribute it for stricter consistency, while AdaBest [31], FedADC [32], and ComFed [33] adaptively calculate local momentum on clients and synchronize each round. Methods like MFL [34] and FedMIM [23] entirely apply momentum on-device to reduce communication. In parallel, momentum-based designs specifically tailored for long-tailed non-IID heterogeneity, such as FedWCM [24, 25], provide an efficient complementary approach. Although momentum mechanisms significantly improve early-stage convergence in non-IID scenarios, they can sometimes hinder late-stage fine-tuning, causing notable performance fluctuations.

### 2.3 SAM-Based Federated Learning

Model generalization is closely tied to finding flatter regions of the loss surface, motivating Sharpness-Aware Minimization (SAM) [35]. SAM actively seeks flatter optima and reduces overfitting risks by perturbing the model around local minima. In the federated setting, works like FedSAM [26], MoFedSAM [26], and FedGAMMA [27] plug SAM optimizers into FedAvg, FedCM, or SCAFFOLD but do not explicitly refine the global flatness search. FedSMOO [28] integrates FedDyn regularization to minimize local-global bias, whereas FedLESAM [36] estimates global perturbations based on local-server discrepancies. More recently, FedGloss [37] extends FedSMOO by leveraging the global pseudo-gradient from the previous round to reduce communication costs, while FedSFA [38] selectively applies SAM perturbations using historical information to lower computational cost. While these approaches enhance generalization, exploring flat minima inevitably slows convergence, and SAM's two backward passes increase computational overhead, further complicating its practical deployment in heterogeneous FL.

# 3 Preliminaries

## 3.1 Federated Learning

Federated Learning (FL) [1] enables multiple clients to train a global model collaboratively while preserving data privacy. The objective is $\min_w F(w) = \sum_{k=1}^K \frac{n_k}{n} F_k(w)$, where $F_k(w)$ is the local loss at client $k$, $n_k$ is client $k$'s data size, and $n = \sum_{k=1}^K n_k$. Parameter $w$ denotes the global model.

## 3.2 Momentum Method

A standard form is: $v_k^r = \alpha\, g_k^r + (1-\alpha)\, \Delta^r$, where $v_k^r$ is the momentum at client $k$ in round $r$, $\alpha$ is the momentum factor, $g_k^r$ is the local gradient, and $\Delta^r$ is the global momentum. Methods like MIME and FedCM employ this scheme to coordinate and stabilize local optimization under non-IID conditions.

## 3.3 SAM Method

Sharpness-Aware Minimization (SAM) [35] improves generalization by identifying flatter regions of the loss function. Its objective is: $\min_w F_{\mathrm{SAM}}(w) = \min_w \max_{\|\delta\|_2 \leq \rho} \left[ \mathbb{E}(L(w+\delta) - L(w)) \right]$. Implementation typically involves two steps: (1) compute $\delta = \rho \frac{\nabla F(w)}{\|\nabla F(w)\|}$ and (2) update model parameters $w$ based on the perturbed gradient $L(w+\delta)$. This forces the model to converge to flatter minima by explicitly considering the worst-case local perturbation.

## 3.4 Motivation

The core challenge of applying SAM in federated learning lies in a fundamental mismatch: SAM computes perturbations based on local data, but aims to find flat minima in the global landscape (Figure 1). Due to data heterogeneity, these local perturbations often fail to accurately reflect the global geometry, thereby limiting the effectiveness of SAM in federated settings.

To resolve this contradiction, existing methods attempt to either reduce the discrepancy between local and global models or improve the estimation of global-aware perturbations using local information. However, as analyzed in `Appendix A`, these approaches still fall short in fully bridging the local-global gap, motivating the need for a more unified and efficient solution. This motivates us to propose a new approach that more effectively integrates SAM with federated optimization.

# 4 Method

This section presents our **FedWMSAM** algorithm, which fuses personalized momentum and SAM for federated optimization. Section 4.1 highlights our methodology and illustrates the three key components (personalized momentum, global perturbation estimation, and dynamic weighting). Section 4.2 then details the implementation, including pseudocode, the computation of personalized momentum $\Delta_r^k$ and dynamic weight $\alpha_r$.

Figure 1: The core of SAMs.

## 4.1 Methodology

**(a) Personalized momentum for local–global discrepancy.** We use the diagram in Figure 2 (a) to illustrate the concept of personalized momentum. In FedCM [22], momentum utilizes the previous gradients to guide the next round of local training, which can effectively accelerate model convergence, as the black dashed line shows. However, it is worth noting that although each client has different data distributions, they share the same momentum. Although momentum mitigates part of the bias introduced by local gradients, the drift caused by data heterogeneity recurs in every communication round and cannot be fully corrected by momentum alone. Based on this, we introduce a correction term $c$ from SCAFFOLD to estimate the bias caused by local data using historical experience. Unlike the original SCAFFOLD [11], which requires uploading local correction terms in each round for global averaging and redistribution, our approach only requires the server to compute the correction term based on the differences in gradients uploaded by clients. This correction term $c_k^r$ in the green dashed line is then aggregated with the momentum $\Delta_r$ to form personalized momentum $\Delta_r^k$, as shown in the red dashed line, effectively saving communication bandwidth.

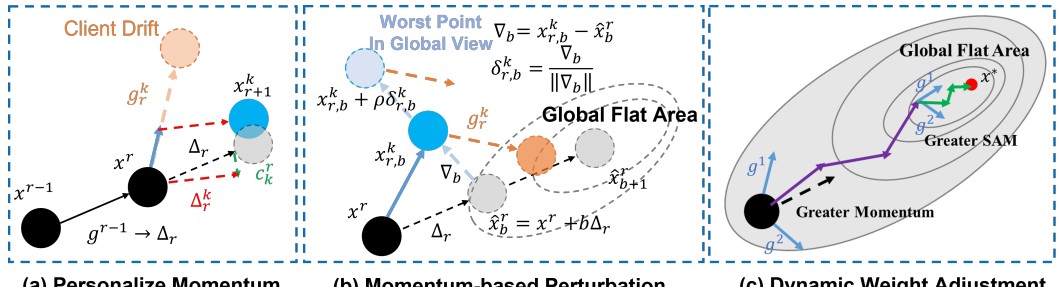

**(a) Personalize Momentum**     **(b) Momentum-based Perturbation**     **(c) Dynamic Weight Adjustment**

Figure 2: FedWMSAM idea: (a) personalized momentum reduces local-global discrepancy; (b) local model vs. momentum-based model difference guides global perturbation estimation; (c) a dynamic weighting adjusts momentum v.s. SAM based on gradient–momentum similarity.

**(b) Global perturbation estimated from momentum.** Performing true SAM in a federated setting is challenging because clients have access only to their local data. Yet they must collaboratively explore the flat region of the global loss landscape. More information about the global loss function is necessary for local SAM. We observe that it is the momentum that carries information about the direction in which the global model is moving. As shown in Figure 2(b), by adding the momentum $\Delta_r$ to the received model $x^r$, we can infer the position of the global model $\hat{x}_b^r$, where the subscript $b$ denotes the current batch. It is important to note that the momentum here is personalized, and the correction term helps eliminate the intrinsic perturbation drift caused by local data distributions.

At each step, we calculate the difference between the current model and the inferred global model, using $\nabla_b$ in light blue dashed line as the perturbation direction, based on the analysis that the direction of deviation from the global model indicates regions of higher loss. By doing this, we eliminate the need for an additional backpropagation step to compute the perturbation while improving the estimation of the global perturbation.

**(c) Dynamic Weighting via Gradient–Momentum Similarity.** Although momentum accelerates early convergence and SAM improves final accuracy, their roles vary throughout the training process. An overly significant momentum can hinder late-stage fine-tuning, while pure SAM is relatively slow, as shown in Figure 3. Inspired by experiments conducted by Andriushchenko et al. [39], which show that switching from ERM to SAM at different epochs can lead to varying test errors, we realized that it is crucial to determine the optimal time to increase the weight of SAM.

We observe that the cosine similarity between clients' directions and the global signal increases early and stabilizes later (see `Appendix B`), so we increase $\alpha_r$ monotonically with the similarity—yielding *early-momentum, late-SAM*. This allows us to rely more on momentum during the early stages and gradually weaken its influence to better explore the global flat region in the final stages. Figure 2 (c) shows how momentum is helpful to speed up initially in the purple line, then gradually yields to SAM for robust final convergence in the green line.

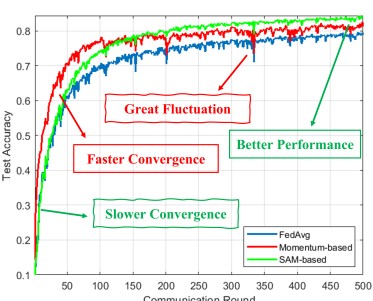

Figure 3: Momentum vs SAM in FL.

### 4.2 Proposed Algorithm

In this section, we describe the components and procedures of the **FedWMSAM** algorithm. First, we provide an overview of the method, followed by detailed explanations of each component. Algorithm 1 summarizes the overall procedure.

#### 4.2.1 Algorithm Overview

At each communication round $r$, the server selects a subset of clients $\mathcal{P}_r$ and computes each client's personalized momentum. The server then broadcasts the global model $x_r$, the personalized momentum $\Delta_r^k$, and the momentum factor $\alpha_r$ to the selected clients. Each client updates its model using the global momentum and performs local updates with SAM perturbation. The momentum

factor is adapted based on cosine similarity, and personalized corrections are refined following the SCAFFOLD-like method to reduce local-global drift. Next, we explain each of these steps in detail.

### 4.2.2 Personalized Momentum Calculation

At each round $r$, the server computes the personalized momentum for each selected client $k$ by:

$$\Delta_r^k = \Delta_r + \frac{\alpha_r}{1 - \alpha_r}\, c_k, \tag{1}$$

where $\Delta_r$ is the global momentum, $\alpha_r$ is the momentum factor, and $c_k$ is the local correction for client $k$. This step ensures the alignment of the global and local updates. The coefficient $\frac{\alpha_r}{1-\alpha_r}$ arises because, during the local updates, the momentum term is combined with the gradient as:

$$v_{b+1,k}^r = \alpha_r\, g_{b,k}^r + (1 - \alpha_r)\, \Delta_r^k. \tag{2}$$

We want the correction term $c_k$ to affect the gradient $g_{b,k}^r$ directly, so the coefficient $\frac{\alpha_r}{1-\alpha_r}$ is used to scale $c_k$ in the global momentum computation. This adjustment ensures that $c_k$ has the same effect on the gradient as it would in the local update, enabling a consistent blend of global and local dynamics in the personalized momentum. This transformation decouples the correction term from momentum, reducing the need to transmit separate vectors for momentum and correction and saving bandwidth.

### 4.2.3 Adaptive Momentum Factor

After each round, we update the momentum factor $\alpha_{r+1}$ based on the cosine similarity between the global momentum and each client's personalized momentum. For efficiency, we compute the similarity using $\mathrm{sim}(\Delta_r, \Delta_r^k)$, which we empirically find to be a good proxy for the gradient–momentum similarity without incurring extra backpropagation (see `Appendix B`). The updated momentum factor is:

$$\hat{\alpha}_{r+1} = \frac{1}{|\mathcal{P}_r|} \sum_{k \in \mathcal{P}_r} \mathrm{sim}(\Delta_r, \Delta_r^k), \tag{3}$$

and the final update is:

$$\alpha_{r+1} = (1 - \lambda)\, \alpha_r + \lambda\, \min\Big(\max\big(\hat{\alpha}_{r+1}, 0.1\big), 0.9\Big), \tag{4}$$

where $\lambda$ controls the speed of adaptation. The choice of the bounds for $\alpha_{r+1}$, specifically within the range $[0.1, 0.9)$, is motivated by several factors. Based on the analysis in [22], a momentum factor of $0.1$ yields the best performance. This setting ensures that the momentum gradually decreases over time, allowing the SAM perturbation to become more prominent in later rounds. Therefore, the upper bound of $0.9$ ensures that momentum remains sufficiently large, maintaining its influence on the calculation of SAM, which relies on the momentum term. The lower bound $0.1$ prevents the momentum weight $1 - \alpha_r$ from vanishing too early, while the upper bound $0.9$ avoids over-reliance on momentum so that SAM can dominate in later rounds. More discussion of this choice of the value can be found in `Appendix B`.

### 4.2.4 Correction Term Updates

Finally, the server updates the personalized correction terms $c_k$ and $c_g$ using a SCAFFOLD-inspired strategy [11]. These updates help reduce local-global drift and refine the correction offsets for each client and are calculated as:

$$c_k^{r+1} = c_k^r - c_g^r - \frac{1}{\eta_l B}\Delta_k^r, \quad c_g^{r+1} = c_g^r + \frac{1}{|\mathcal{P}_r|} \sum_{k \in \mathcal{P}_r} (c_k^{r+1} - c_k^r). \tag{5}$$

### 4.3 Convergence Analysis

We provide a non-IID convergence guarantee for **FedWMSAM**, with a rate of $\widetilde{\mathcal{O}}\big(\sqrt{L\Delta\,\sigma_\rho^2/(SKR)} + \frac{L\Delta}{R}\big(1 + \frac{N^{2/3}}{S}\big)\big)$. Our analysis builds upon the SCAFFOLD-M framework [40], and explicitly accounts for the perturbation-induced variance $\sigma_\rho^2 = \sigma^2 + (L\rho)^2$, with two key extensions: an adaptive personalized momentum term and the integration of SAM. Notably, we show that the variance induced by SAM is bounded by the perturbation strength $\rho$. The complete proof and comparisons with related convergence rates are presented in `Appendix A`.

**Algorithm 1** FedWMSAM

**Require:** Initial model $x_0$, global momentum $\Delta_0$, correctors $c_g, c_k$, momentum factor $\alpha_0 = 0.1$, learning rates $\eta_l, \eta_g$, perturbation magnitude $\rho$, communication rounds $R$, local iters $B$

1: **for** $r = 0$ **to** $R - 1$ **do**
2:      **for each** client $k \in \mathcal{P}_r$ **in parallel do**
3:          **Compute** $\Delta_r^k$ using Eq. (1)
4:          $\Delta_k^r = \textbf{CLIENTUPDATE}(x_r, \Delta_r^k, \alpha_r)$
5:      **end for**
6:      $\Delta_{r+1} = \frac{1}{\eta_l |\mathcal{P}_r|} \sum_{k \in \mathcal{P}_r} \Delta_k^r$
7:      $x_{r+1} = x_r - \eta_g \Delta_{r+1}$
8:      **Compute** $\alpha_{r+1}$ using Eq. (3) and (4)
9:      **Update** $c_g, c_k$ using Eq. (5)
10: **end for**
**Ensure:** Return updated model $x_R$

11: **CLIENTUPDATE**$(x_r, \Delta_r^k, \alpha_r)$:
12: $x_{0,k}^r = x_r$
13: **for** $b = 0$ **to** $B - 1$ **do**
14:      $\delta_{b+1,k}^r = (x_r + b \, \Delta_r^k) - x_{b,k}^r$
15:      $g_{b,k}^r = \nabla\mathcal{L}(x_{b,k}^r + \rho \, \delta_{b+1,k}^r / \|\delta_{b+1,k}^r\|)$
16:      $v_{b+1,k}^r = \alpha_r \, g_{b,k}^r + (1 - \alpha_r) \, \Delta_r^k$
17:      $x_{b+1,k}^r = x_{b,k}^r - \eta_l \, v_{b+1,k}^r$
18: **end for**
19: $\Delta_k^r = x_{B,k}^r - x_r$
**Ensure:** Return Client update $\Delta_k^r$

## 5 Experiment

### 5.1 Experimental Setups

We propose FedWMSAM and compare it with existing SOTA federated SAM methods, including FedSAM [26], MoFedSAM [26], FedGAMMA [27], and FedSMOO [28], as well as FedLESAM [36] and its two variants, FedLESAM-S and FedLESAM-D. Since our method incorporates momentum and correction terms, we also compare it with classical federated optimization baselines such as FedAvg [1], FedCM [22], and SCAFFOLD [11]. By default, we set $p_{k,c} \sim \text{Dir}(\beta)$, where $p_{k,c}$ denotes the class distribution of client $k$ over class $c$, and $\beta = 0.1$. The main experiments are conducted with 100 clients, 10% participation per round, a batch size of 50, a local learning rate $\eta_l = 0.1$, a global learning rate $\eta_g = 1$, and five local epochs, running for 500 communication rounds. For Fashion-MNIST [41], we use a Multi-Layer Perceptron (MLP) architecture. For CIFAR-10 [42], we use ResNet-18 [43] as the backbone, ResNet-34 [43] for CIFAR-100 [42], and ResNet-50 [43] for *OfficeHome*. Each domain in *OfficeHome* is divided into one client with 10% data sample rate and 100% active ratio. The perturbation magnitude $\rho$ is set to 0.01 for FedSAM, FedGAMMA, FedLESAM, and our FedWMSAM, while FedSMOO and MoFedSAM use $\rho = 0.1$ by default. Additional experimental settings are detailed in the corresponding figures and tables. All experiments were implemented in PyTorch and conducted on a workstation with four NVIDIA GeForce RTX 3090 GPUs.

### 5.2 Overall Performance Evaluation

Table 1: Performance comparison of SOTA methods under Dirichlet and Pathological splits after 500 Rounds in different datasets.

| Method | Fashion-MNIST | | | | CIFAR-10 | | | | CIFAR-100 | | | |
|---|---|---|---|---|---|---|---|---|---|---|---|---|
| #Partition | Dirichlet | | Pathological | | Dirichlet | | Pathological | | Dirichlet | | Pathological | |
| #Coefficient | $\beta = 0.6$ | $\beta = 0.1$ | $\gamma = 6$ | $\gamma = 3$ | $\beta = 0.6$ | $\beta = 0.1$ | $\gamma = 6$ | $\gamma = 3$ | $\beta = 0.6$ | $\beta = 0.1$ | $\gamma = 20$ | $\gamma = 10$ |
| FedAvg | 0.8684 | 0.8226 | 0.8625 | 0.8150 | 0.7886 | 0.7005 | 0.7873 | 0.6426 | 0.3917 | 0.3815 | 0.3968 | 0.3631 |
| FedCM | 0.8283 | 0.7333 | 0.8047 | 0.6630 | 0.8126 | 0.7229 | 0.8167 | 0.7025 | 0.4635 | 0.4290 | 0.4394 | 0.3940 |
| SCAFFOLD | 0.8789 | 0.8351 | 0.8785 | 0.8311 | 0.8232 | 0.7428 | 0.8179 | 0.6786 | 0.4855 | 0.4437 | 0.4647 | 0.4133 |
| FedSAM | 0.8683 | 0.8261 | 0.8673 | 0.8045 | 0.7963 | 0.6963 | 0.7908 | 0.6503 | 0.4083 | 0.3790 | 0.3933 | 0.3553 |
| MoFedSAM | 0.8278 | 0.7489 | 0.8141 | 0.6822 | 0.8339 | 0.7386 | 0.8334 | 0.7327 | 0.4859 | 0.4472 | 0.4619 | 0.4279 |
| FedGAMMA | 0.8708 | 0.8298 | 0.8716 | 0.8303 | 0.8292 | 0.7218 | 0.8043 | 0.6105 | 0.4837 | 0.4474 | 0.1739 | 0.0198 |
| FedSMOO | **0.8846** | 0.8337 | 0.8745 | 0.8296 | **0.8410** | 0.7507 | 0.8382 | 0.7099 | 0.3225 | 0.2987 | 0.4620 | 0.3006 |
| FedLESAM(-S/-D) | 0.8689 | 0.8375 | 0.8732 | 0.8209 | 0.8165 | 0.7284 | 0.8127 | 0.6381 | 0.4260 | 0.4114 | 0.4298 | 0.3914 |
| **FedWMSAM (ours)** | 0.8756 | **0.8464** | **0.8805** | **0.8531** | 0.8356 | **0.7664** | **0.8443** | **0.7446** | **0.4908** | **0.4646** | **0.4786** | **0.4383** |

Note 1: We report the best accuracy among FedLESAM, FedLESAM-S, and FedLESAM-D in one row.
Note 2: $\gamma$ represents the number of classes allocated to each client in the pathological distribution.

**Accuracy Evaluation.** Table 1 summarizes results across three datasets and twelve heterogeneity settings. **FedWMSAM** is best or second-best in most cases, with the advantage most pronounced under stronger non-IID settings ($\beta$=0.1 and pathological splits). Specifically, on CIFAR-10/100 at

$\beta$=0.1 we achieve **76.64%/46.46%**, improving over baseline FedAvg by +6.59/+8.31 points. Similarly, under scenarios where $\gamma = 3$ for CIFAR-10 and $\gamma = 10$ for CIFAR-100, FedWMSAM attains accuracies of 74.46% and 43.83%, surpassing FedAvg by +10.2 and +7.52 points, respectively.

The results show that on easier splits, such as CIFAR-10 at $\beta$=0.6, the difference compared to the strongest SAM baseline is smaller. These gains concentrate where the local–global curvature misalignment is more severe, supporting our momentum-guided global perturbation design. Moreover, improvements emerge precisely where our non-IID bound predicts larger variance terms (low sampling $S$ and greater heterogeneity). Under stronger non-IID conditions, our improvements become more significant, indicating that alignment, rather than sheer capacity, drives these enhancements.

Table 2: Accuracy on OfficeHome target domains after 500 rounds (10% sample, 100% active).

| Method | Art | Clipart | Product | Real World |
|---|---|---|---|---|
| FedAvg | 0.9909 | 0.9569 | 0.9725 | 0.9633 |
| FedCM | 0.9316 | 0.8013 | 0.8783 | 0.8411 |
| SCAFFOLD | 0.9934 | 0.9610 | 0.9745 | **0.9749** |
| FedSAM | 0.9851 | 0.9402 | 0.9576 | 0.9685 |
| MoFedSAM | 0.9921 | 0.9458 | 0.9653 | 0.9566 |
| FedGamma | 0.9934 | 0.9557 | 0.9758 | 0.9605 |
| FedSMOO | 0.9868 | 0.9563 | 0.9753 | 0.9629 |
| FedLESAM | 0.9930 | 0.9626 | 0.9783 | 0.9713 |
| FedWMSAM | **0.9942** | **0.9650** | **0.9790** | 0.9717 |

To demonstrate the adaptability of our method to the heterogeneity of real-world data, we conducted experiments on the *OfficeHome* dataset. As shown in Table 2, **FedWMSAM** is best on 3/4 domains (Art/Clipart/Product) and achieves the best average, slightly trailing SCAFFOLD on Real-World. This suggests that aligning local updates to a global direction transfers across domains, complementing variance-reduction methods under specific domains.

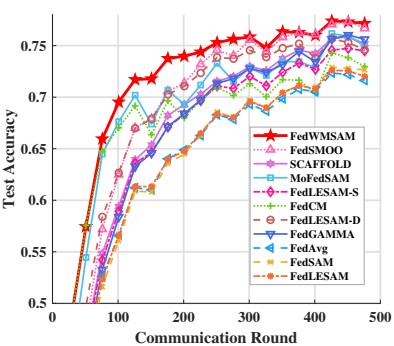 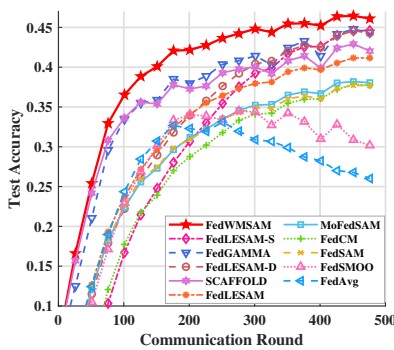

Figure 4: Performance comparison on CIFAR-10 (left) and CIFAR-100 (right).

**Convergence Efficiency.** We compare the convergence rate of FedWMSAM in Figure 4 and Table 3. The results indicate that Fed-WMSAM achieves the target accuracy significantly faster than other methods and dramatically reduces training time compared to classical SAM-based methods. In detail, **FedWMSAM** matches the fastest early-stage baselines at 0.70 (97 rounds, on par with FedCM/MoFedSAM) and then surpasses them at higher targets (**114**/153/241/356 rounds to 0.72/0.74/0.76/0.78). Per-round client time is near FedAvg/FedLESAM (15.03s vs. 14.57–16.96s) and markedly lower than double-backprop SAM-family methods (26.9–29.8s).

Table 3: Rounds to reach different accuracy levels and client computation time.

| Method | 0.7 | 0.72 | 0.74 | 0.76 | 0.78 | Time(s) |
|---|---|---|---|---|---|---|
| FedAvg | 254 | 348 | 432 | - | - | **14.57** |
| FedCM | **97** | 132 | 255 | 426 | - | 14.67 |
| SCAFFOLD | 189 | 241 | 301 | 376 | - | 17.44 |
| FedSAM | 247 | 303 | 403 | - | - | 26.90 |
| MoFedSAM | **97** | 132 | 169 | 255 | 426 | 29.73 |
| FedGAMMA | 208 | 241 | 300 | 374 | - | 29.72 |
| FedSMOO | 134 | 167 | 201 | 255 | 382 | 29.77 |
| FedLESAM | 241 | 303 | 433 | - | - | 14.66 |
| FedLESAM-D | 149 | 187 | 211 | 255 | - | 16.96 |
| FedLESAM-S | 205 | 247 | 313 | 432 | - | 16.71 |
| FedWMSAM | **97** | **114** | **153** | **241** | **356** | 15.03 |

Mechanistically, the *cosine-similarity adaptive coupling* induces an early-momentum, late-SAM two-phase behavior that dampens *momentum-echo oscillation*, yielding the observed *fast-then-flat* trajectories.

**Generalization Illustration.** To intuitively illustrate the generalization ability of our method, we present the t-SNE visualization of the global models trained by four FL algorithms on CIFAR-10: FedAvg, FedSAM, MoFedSAM, and our FedWMSAM. As shown in Figure 5, 13, and 12,

FedWMSAM achieves better class separation and more compact clusters, consistent with flatter minima and reduced inter-client discrepancy, indicating improved generalization under non-IID settings, echoing the *fast & flat* objective in our theory. More results and discussions are provided in `Appendix C`.

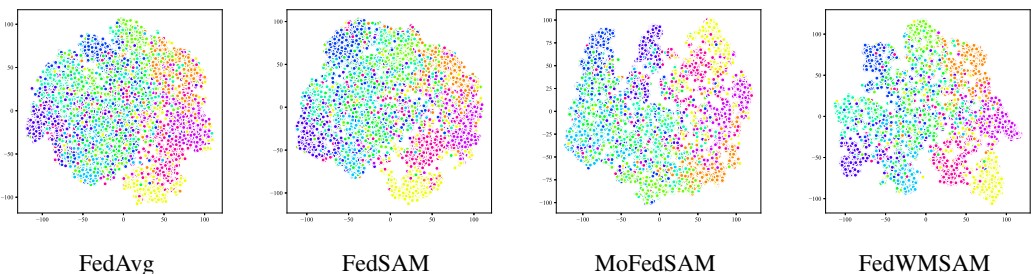

Figure 5: t-SNE visualization results of client embeddings using selected FL algorithms.

## 5.3 Sensitivity Analysis

To validate the adaptability of our method to various experimental settings, we study sensitivity along three axes: (*i*) local epochs, (*ii*) number of clients, and (*iii*) client sampling rate, comparing it with FedAvg and SAM-family baselines ( MoFedSAM, FedSAM, and FedSMOO).

Table 4: Comparison under different local epochs.

| Method | Epoch 1 | Epoch 5 | Epoch 10 | Epoch 20 |
|---|---|---|---|---|
| FedAvg | 0.6003 | 0.7005 | 0.6988 | 0.6879 |
| MoFedSAM | 0.7237 | 0.7386 | 0.6997 | 0.6776 |
| FedSAM | 0.5515 | 0.6963 | 0.6862 | 0.6903 |
| FedSMOO | **0.7888** | 0.7507 | 0.7538 | 0.7472 |
| FedWMSAM (Ours) | 0.7484 | **0.7664** | **0.7662** | **0.7515** |

**Local epochs.** Table 4 shows the comparison with different local epoch settings. Unlike other methods, **FedWMSAM** maintains the highest accuracy across most settings for $K \in \{5, 10, 20\}$ and shows no clear drop in accuracy as the number of local epochs increases, indicating stable optimization with larger local computations for dynamically estimating global perturbations.

**Number of clients.** Figure 6 (left) illustrates the comparison under different client numbers. **FedWMSAM** is best or on par across the range and consistently leads once the population is $\geq 75$. At minimal populations, *MoFedSAM* can be unstable (e.g., 0.2161 at 20 clients), while *FedSMOO* is competitive only in the mid range (e.g., 50 clients).

**Client sampling rate.** Figure 6 (right) presents the comparison under varying sampling rates. **FedWMSAM** outperforms others across most sampling rates, even when participation is sparse; as the rate increases (e.g., $\geq 80\%$), all methods plateau and the gap narrows. Accuracy saturates beyond 20%. Gains are most pronounced when participation is sparse or heterogeneity is stronger (few clients / low sampling), consistent with our momentum-guided global perturbation and cosine-based adaptive coupling.

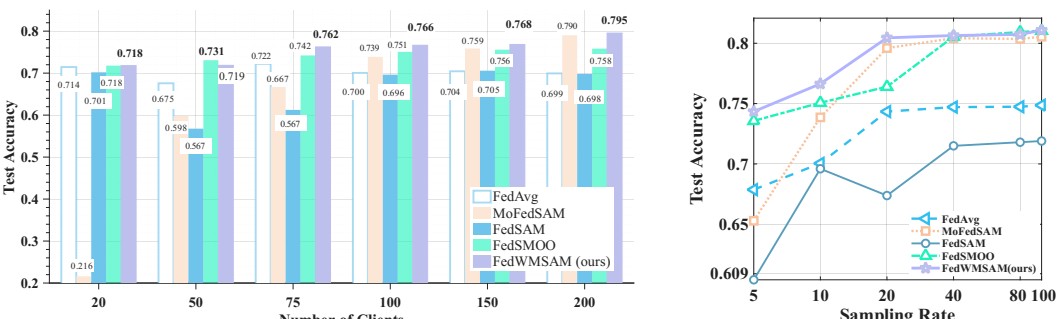

Figure 6: Performance comparison across different numbers of clients and sampling rates.

## 5.4 Ablation Study

We conducted ablation studies on our method and the perturbation coefficient $\rho$ to validate its systematic design and parameter choices. We ablate three modules: *personalized momentum* (Mom.), *momentum-guided SAM* (SAM, single backprop), and the *cosine-adaptive weight* (Weighted; $\alpha_r$). The bottom row ($\times/\times/\times$) corresponds to *FedCM* (no personalized momentum, no SAM, no adaptive coupling), and the "Imp." is computed *over FedCM*.

Table 5: Ablation of key modules in FedWMSAM.

| Mom. | SAM | Weighted | Acc. | Imp. |
|------|-----|----------|------|------|
| ✓ | ✓ | ✓ | 0.7664 | 4.35% |
| ✓ | ✓ | × | 0.7556 | 3.27% |
| × | ✓ | ✓ | 0.7265 | 0.36% |
| ✓ | × | ✓ | 0.7326 | 0.97% |
| ✓ | × | × | 0.7478 | 2.49% |
| × | ✓ | × | 0.7430 | 2.01% |
| × | × | × | 0.7229 | - |

Table 5 presents the performance of different component combinations. The personalized momentum primarily reduces local-global bias, which accounts for the significant gains observed with the Mom.-only approach.

The cosine-adaptive rule serves as a *data-driven damping* mechanism. When both Mom.& SAM exist, it facilitates an early-Mon. and late-SAM schedule. This combination helps suppress momentum echo and enhances the final performance plateau. In contrast, when SAM is not present, the same gate may diminish the momentum's acceleration and bring only marginal benefit. Overall, the three modules work together complementarily: **Mom.** fixes bias, **SAM** supplies flatness, and **Weighted** harmonizes the two.

We further vary the perturbation magnitude $\rho$. Table 6 shows that accuracy peaks around $\rho$=0.01, and **FedWMSAM** degrades gracefully as $\rho$ increases (0.7664 $\rightarrow$ 0.7563 $\rightarrow$ 0.7244 $\rightarrow$ 0.5905 from 0.01 $\rightarrow$ 0.05 $\rightarrow$ 0.1$\rightarrow$ 0.5), whereas *MoFedSAM* collapses at large $\rho$ (0.7102 $\rightarrow$ 0.5562 $\rightarrow$ 0.1000).

Table 6: Ablation study results of $\rho$.

| Method $\rho$ | 0.005 | 0.01 | 0.05 | 0.1 | 0.5 |
|---------------|-------|------|------|-----|-----|
| FedAvg | 0.7005 | 0.7005 | 0.7005 | 0.7005 | 0.7005 |
| MoFedSAM | 0.7355 | 0.7386 | 0.7102 | 0.5562 | 0.1000 |
| FedWMSAM (ours) | **0.7659** | **0.7664** | **0.7563** | **0.7244** | **0.5905** |

Our momentum-guided perturbation aligns the SAM direction with the global geometry. The cosine-adaptive weight reduces SAM's contribution when there is misalignment or noise. This results in a milder *effective* perturbation during noisy rounds, which corresponds to the bound's dependence on $\sigma_\rho^2 = \sigma^2 + (L\rho)^2$. As $\rho$ increases, the noise term also rises, but the adaptive gate mitigates its effect, preventing the catastrophic failure observed in *MoFedSAM*.

## 5.5 Supplementary Experiments

We provide additional experiments in `Appendix B` and `Appendix C`. `Appendix B` includes studies on the cosine-based computation of $\alpha_r$ and the effect of decay coefficient $\lambda$. `Appendix C` presents additional results and visualizations across datasets and partitioning settings, offering more detailed comparisons that support the robustness of FedWMSAM.

## 6 Conclusion

In this paper, we addressed why naively combining momentum and SAM under non-IID FL underperforms by *identifying and formalizing* two failure modes: *local–global curvature misalignment* and *momentum-echo oscillation*. We introduced **FedWMSAM**, which (i) builds a *momentum-guided global perturbation* to align local SAM directions with the global descent geometry (single backprop), and (ii) *couples* momentum and SAM via a *cosine-similarity adaptive rule* that yields an early-momentum / late-SAM two-phase schedule. On the theory side, we gave a non-IID convergence bound that *explicitly models the perturbation-induced variance* $\sigma_\rho^2 = \sigma^2 + (L\rho)^2$ and its dependence on $(S, K, R, N)$. Empirically, across three datasets and twelve heterogeneity settings, FedWMSAM is best or on-par in most cases, with the *largest gains in strong non-IID*, and it reaches target accuracies in *fewer rounds* while maintaining *near-FedAvg per-round cost*, realizing the intended *fast-and-flat* optimization.

## Acknowledgments

This work was supported in part by the Guangdong Provincial Key Laboratory of Integrated Communication, Sensing and Computation for Ubiquitous Internet of Things (No. 2023B1212010007); the National Natural Science Foundation of China (NSFC) under Grants 62372307, 62472366, and U2001207; the Guangdong Natural Science Foundation under Grant 2024A1515011691; the Project of DEGP under Grants 2023KCXTD042 and 2024GCZX003; the Shenzhen Science and Technology Foundation under Grants JCYJ20230808105906014 and ZDSYS20190902092853047; the Shenzhen Science and Technology Program under Grant RCYX20231211090129039; and the 111 Center (No. D25008).

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

# A  Convergence Proof and Analysis

## A.1  Proof of Convergence for FedWMSAM

### A.1.1  Notations and Definitions

Let $F_0 = \emptyset$ and $F_i^{r,k} := \sigma\left(\{x_i^{r,j}\}_{0 \leq j \leq k} \cup F_r\right)$ and $F_{r+1} := \sigma\left(\bigcup_i F_i^{r,K}\right)$ for all $r \geq 0$, where $\sigma(\cdot)$ indicates the $\sigma$-algebra. Let $\mathbb{E}_r[\cdot] := \mathbb{E}[\cdot \mid F_r]$ be the expectation, conditioned on the filtration $F_r$, with respect to the random variables $\{S_r, \{\xi_i^{r,k}\}_{1 \leq i \leq N, 0 \leq k < K}\}$ in the $r$-th iteration. We also use $\mathbb{E}[\cdot]$ to denote the global expectation over all randomness in algorithms.

Throughout the proofs, we use $\sum_i$ to represent the sum over $i \in \{1, \ldots, N\}$, while $\sum_{i \in S_r}$ denotes the sum over $i \in S_r$. Similarly, we use $\sum_k$ to represent the sum over $k \in \{0, \ldots, K-1\}$. For all $r \geq 0$, we define the following auxiliary variables to facilitate proofs:

$$E_r := \mathbb{E}[\|\nabla f(x^r) - g^{r+1}\|^2],$$

$$U_r := \frac{1}{NK} \sum_i \sum_k \mathbb{E}[\|x_i^{r,k} + \delta_i^{r,k} - x^r\|^2],$$

$$\zeta_i^{r,k} := \mathbb{E}[x_i^{r,k+1} - x_i^{r,k} + \delta_i^{r,k} \mid F_i^{r,k}],$$

$$\Xi_r := \frac{1}{N} \sum_{i=1}^N \mathbb{E}[\|\zeta_i^{r,0}\|^2],$$

$$V_r := \frac{1}{N} \sum_{i=1}^N \mathbb{E}[\|c_i^r - \nabla f_i(x^{r-1})\|^2].$$

Throughout the appendix, we let $\Delta := f(x^0) - f^*$, $G_0 := \frac{1}{N} \sum_i \|\nabla f_i(x^0)\|^2$, $x^{-1} := x^0$, and $E_{-1} := \mathbb{E}[\|\nabla f(x^0) - g^0\|^2]$. We will use the following foundational lemma for all our algorithms.

**Assumption 1** (Standard Smoothness). *Each local objective $f_i$ is $L$-smooth, i.e., $\|\nabla f_i(x) - \nabla f_i(y)\| \leq L\|x - y\|$, for all $x, y \in \mathbb{R}^d$ and $1 \leq i \leq N$.*

**Assumption 2** (Stochastic Gradient). *There exists $\sigma \geq 0$ such that for any $x \in \mathbb{R}^d$ and $1 \leq i \leq N$, $\mathbb{E}_{\xi_i}[\nabla F(x; \xi_i)] = \nabla f_i(x)$ and $\mathbb{E}_{\xi_i}[\|\nabla F(x; \xi_i) - \nabla f_i(x)\|^2] \leq \sigma^2$, where $\xi_i \overset{iid}{\sim} D_i$.*

### A.1.2  Preliminary Lemmas

**Lemma 1.** *Under Assumption 1, if $\gamma L \leq \frac{1}{24}$, the following holds for all $r \geq 0$:*

$$\mathbb{E}[f(x^{r+1})] \leq \mathbb{E}[f(x^r)] - \frac{11\gamma}{24}\mathbb{E}[\|\nabla f(x^r)\|^2] + \frac{13\gamma}{24}\mathcal{E}_r.$$

*Proof.* Since $f$ is $L$-smooth, we have

$$f(x^{r+1}) \leq f(x^r) + \langle \nabla f(x^r), x^{r+1} - x^r \rangle + \frac{L}{2}\|x^{r+1} - x^r\|^2,$$

$$= f(x^r) - \gamma\|\nabla f(x^r)\|^2 + \gamma\langle \nabla f(x^r), \nabla f(x^r) - g^{r+1}\rangle + \frac{L\gamma^2}{2}\|g^{r+1}\|^2.$$

Since $x^{r+1} = x^r - \gamma g^{r+1}$, using Young's inequality, we further have

$$f(x^{r+1}) \leq f(x^r) - \frac{\gamma}{2}\|\nabla f(x^r)\|^2 + \frac{\gamma}{2}\|\nabla f(x^r) - g^{r+1}\|^2 + L\gamma^2(\|\nabla f(x^r)\|^2 + \|\nabla f(x^r) - g^{r+1}\|^2),$$

$$\leq f(x^r) - \frac{11\gamma}{24}\|\nabla f(x^r)\|^2 + \frac{13\gamma}{24}\|\nabla f(x^r) - g^{r+1}\|^2,$$

where the last inequality is due to $\gamma L \leq \frac{1}{24}$. Taking the global expectation completes the proof. $\square$

**Lemma 2** ( [11]). *Suppose $\{X_1, \cdots, X_\tau\} \subset \mathbb{R}^d$ are random variables that are potentially dependent. If their marginal means and variances satisfy $\mathbb{E}[X_i] = \mu_i$ and $\mathbb{E}[\|X_i - \mu_i\|^2] \leq \sigma^2$, then it holds that*

$$\mathbb{E}\left[\left\|\sum_{i=1}^\tau X_i\right\|^2\right] \leq \left\|\sum_{i=1}^\tau \mu_i\right\|^2 + \tau^2 \sigma^2.$$

*If they are correlated in the Markov way such that $\mathbb{E}[X_i \mid X_{i-1}, \cdots, X_1] = \mu_i$ and $\mathbb{E}[\|X_i - \mu_i\|^2 \mid \mu_i] \leq \sigma^2$, i.e., the variables $\{X_i - \mu_i\}$ form a martingale. Then the following tighter bound holds:*

$$\mathbb{E}\left[\left\|\sum_{i=1}^\tau X_i\right\|^2\right] \leq 2\mathbb{E}\left[\left\|\sum_{i=1}^\tau \mu_i\right\|^2\right] + 2\tau\sigma^2.$$

**Lemma 3.** *Given vectors $v_1, \cdots, v_N \in \mathbb{R}^d$ and $\bar{v} = \frac{1}{N} \sum_{i=1}^N v_i$, if we sample $S \subset \{1, \cdots, N\}$ uniformly randomly such that $|S| = S$, then it holds that*

$$\mathbb{E}\left[\left\|\frac{1}{S}\sum_{i \in S} v_i\right\|^2\right] = \|\bar{v}\|^2 + \frac{N-S}{S(N-1)}\frac{1}{N}\sum_{i=1}^N \|v_i - \bar{v}\|^2.$$

*Proof.* Letting $\mathbb{1}\{i \in S\}$ be the indicator for the event $i \in S_r$, we prove this lemma by direct calculation as follows:

$$\mathbb{E}\left[\left\|\frac{1}{S}\sum_{i \in S} v_i\right\|^2\right] = \mathbb{E}\left[\left\|\frac{1}{S}\sum_{i=1}^N v_i \mathbb{1}\{i \in S\}\right\|^2\right]$$

$$= \frac{1}{S^2}\mathbb{E}\left[\sum_i \|v_i\|^2 \mathbb{1}\{i \in S\} + 2\sum_{i<j} v_i^\top v_j \mathbb{1}\{i, j \in S\}\right]$$

$$= \frac{1}{SN}\sum_{i=1}^N \|v_i\|^2 + \frac{1}{S^2}\frac{S(S-1)}{N(N-1)} \cdot 2\sum_{i<j} v_i^\top v_j$$

$$= \frac{1}{SN}\sum_{i=1}^N \|v_i\|^2 + \frac{1}{S^2}\frac{S(S-1)}{N(N-1)}\left(\left\|\sum_{i=1}^N v_i\right\|^2 - \sum_{i=1}^N \|v_i\|^2\right)$$

$$= \frac{N-S}{S(N-1)}\frac{1}{N}\sum_{i=1}^N \|v_i\|^2 + \frac{N(S-1)}{S(N-1)}\|\bar{v}\|^2$$

$$= \frac{N-S}{S(N-1)}\frac{1}{N}\sum_{i=1}^N \|v_i - \bar{v}\|^2 + \|\bar{v}\|^2.$$

$\square$

**Lemma 4** (Perturbation-Induced Gradient Variance). *Suppose each local objective $f_i$ is $L$-smooth (Assumption 1), and the stochastic gradient $\nabla F(x; \xi_i)$ is unbiased with variance at most $\sigma^2$ (Assumption 2). Then for any $x \in \mathbb{R}^d$, any client $i$, and any perturbation vector $\delta$ with $\|\delta\| \leq \rho$, we have*

$$\mathbb{E}_{\xi_i}\|\nabla F(x + \delta; \xi_i) - \nabla f_i(x)\|^2 \leq \sigma^2 + (L\rho)^2.$$

*Proof.* By definition, the random gradient can be decomposed as

$$\nabla F(x + \delta; \xi_i) - \nabla f_i(x) = \underbrace{\left(\nabla F(x + \delta; \xi_i) - \nabla f_i(x + \delta)\right)}_{\text{(A) stochastic deviation}} + \underbrace{\left(\nabla f_i(x + \delta) - \nabla f_i(x)\right)}_{\text{(B) Lipschitz shift}}.$$

We denote these two terms by $A$ and $B$, respectively. Then

$$\|\nabla F(x + \delta; \xi_i) - \nabla f_i(x)\|^2 = \|A + B\|^2 = \|A\|^2 + \|B\|^2 + 2\langle A, B\rangle.$$

From Assumption 2 (Stochastic Gradient), we know

$$\mathbb{E}_{\xi_i}[A] = \mathbb{E}_{\xi_i}\big[\nabla F(x+\delta;\xi_i) - \nabla f_i(x+\delta)\big] = 0,$$

and

$$\mathbb{E}_{\xi_i}\big[\|A\|^2\big] \leq \sigma^2.$$

By Assumption 1 ($L$-smoothness), we have

$$\|\nabla f_i(x+\delta) - \nabla f_i(x)\| \leq L\,\|\delta\| \leq L\rho.$$

Hence,

$$\|B\|^2 \leq (L\rho)^2.$$

Since $B$ is deterministic w.r.t. $\xi_i$, we have $\mathbb{E}_{\xi_i}[A] = 0$, so

$$\mathbb{E}_{\xi_i}[\langle A, B\rangle] = \langle \mathbb{E}_{\xi_i}[A], B\rangle = 0.$$

Thus in expectation,

$$\mathbb{E}_{\xi_i}\big[\|A+B\|^2\big] = \mathbb{E}_{\xi_i}\big[\|A\|^2\big] + \|B\|^2 + 2\,\mathbb{E}_{\xi_i}\langle A, B\rangle \leq \sigma^2 + (L\rho)^2.$$

Putting it all together gives

$$\mathbb{E}_{\xi_i}\Big[\|\nabla F(x+\delta;\xi_i) - \nabla f_i(x)\|^2\Big] \leq \sigma^2 + (L\rho)^2,$$

as claimed. $\qquad\square$

### A.1.3 Proof of FedWMSAM

Following the sketch of SCAFFOLD-M [40], we proceed to prove the convergence of FedWMSAM. We begin by considering the following lemma:

**Lemma 5.** *If $\gamma L \leq \frac{1}{2\alpha_r}$, the following holds for $r \geq 1$:*

$$\mathcal{E}_r \leq \left(1 - \frac{8\alpha_r}{9}\right)\mathcal{E}_{r-1} + 16\alpha_r(\gamma L)^2 \mathbb{E}[\|\nabla f(x^{r-1})\|^2] + \frac{4\alpha_r^2\sigma_\rho^2}{SK} + 10\alpha_r L^2 U_r + 6\alpha_r^2\frac{N-S}{S(N-1)}V_r,$$

*where $\sigma_\rho^2$ denotes $\sigma^2 + (L\rho)^2$. In addition,*

$$\mathcal{E}_0 \leq (1-\alpha_r)\mathcal{E}_{-1} + \frac{4\alpha_r^2\sigma_\rho^2}{SK} + 8\alpha_r L^2 U_0 + 4\alpha_r^2\frac{N-S}{S(N-1)}V_0.$$

*Proof.* According to the definition of personalized momentum in FedWMSAM, denoting $g_r^i$ as the personalized momentum of client $i$ in round $r$, we have

$$\mathcal{E}_r = \mathbb{E}[\|\nabla f(x^r) - g_{r+1}\|^2]$$

$$= \mathbb{E}\left[\|(1-\alpha_r)(\nabla f(x^r) - g_r^i) + \alpha_r\left(\nabla f(x^r) - \frac{1}{K}\sum_i\sum_k \nabla F(x_{r,k}^i;\xi_{r,k}^i)\right)\|^2\right]$$

$$= \mathbb{E}\left[\|(1-\alpha_r)(\nabla f(x^r) - (g_r + \frac{\alpha_r}{1-\alpha_r}(c_i^r - c^r)) + \alpha_r\left(\nabla f(x^r) - \frac{1}{K}\sum_i\sum_k \nabla F(x_{r,k}^i;\xi_{r,k}^i)\right)\|^2\right]$$

$$= \mathbb{E}\left[\|(1-\alpha_r)(\nabla f(x^r) - g_r) + \alpha_r\left(\nabla f(x^r) - \frac{1}{K}\sum_i\sum_k \nabla F(x_{r,k}^i;\xi_{r,k}^i) - (c_i^r - c^r)\right)\|^2\right].$$

Note that $\frac{1}{N}\sum_{i=1}^N c_i^r = c^r$ holds for any $r \geq 0$. Using Lemma 3, $\mathcal{E}_r$ can be expressed as:

$$\mathcal{E}_r = \mathbb{E}\left[\left\|\nabla f(x^r) - \frac{1}{NK}\sum_{i,k} g_i^{r,k}\right\|^2\right] + \frac{\alpha_r^2(N-S)}{S(N-1)}\frac{1}{N}\sum_{i=1}^N \mathbb{E}\left[\left\|\frac{1}{K}\sum_k g_i^{r,k} - \frac{1}{NK}\sum_{j,k} g_j^{r,k}\right\|^2\right].$$

To simplify, we define:

$$\Lambda_1 := \mathbb{E}\left[\left\|(1-\alpha_r)\left(\nabla f(x^r) - g^r\right) + \alpha_r \left(\frac{1}{NK}\sum_{i,k}\nabla F(x_i^{r,k} + \delta_i^{r,k}; \xi_i^{r,k}) - \nabla f(x^r)\right)\right\|^2\right],$$

$$\Lambda_2 := \frac{1}{N}\sum_{i=1}^{N}\mathbb{E}\left[\left\|\frac{1}{K}\sum_k \nabla F(x_i^{r,k} + \delta_i^{r,k}; \xi_i^{r,k}) - \frac{1}{NK}\sum_{j,k}\nabla F(x_j^{r,k} + \delta_j^{r,k}; \xi_j^{r,k}) - (c_i^r - c^r)\right\|^2\right].$$

Thus, $\mathcal{E}_r$ can be rewritten as:

$$\mathcal{E}_r = \Lambda_1 + \alpha_r^2 \frac{N-S}{S(N-1)}\Lambda_2.$$

For $r \geq 1$, expand $\Lambda_1$ we can get,

$$\Lambda_1 = \mathbb{E}\left[\|(1-\alpha_r)(\nabla f(x^r) - g^r)\|^2\right] + \alpha_r^2 \mathbb{E}\left[\left\|\nabla f(x^r) - \frac{1}{NK}\sum_{i,k}\nabla F(x_i^{r,k} + \delta_i^{r,k}; \xi_i^{r,k})\right\|^2\right]$$

$$+ 2\alpha_r \mathbb{E}\left[\langle(1-\alpha_r)(\nabla f(x^r) - g^r), \nabla f(x^r) - \frac{1}{NK}\sum_{i,k}\nabla F(x_i^{r,k} + \delta_i^{r,k})\rangle\right].$$

Note that $\{\nabla F(x_i^{r,k} + \delta_i^{r,k}; \xi_i^{r,k})\}_{0 \leq k < K}$ are sequentially correlated. Applying the AM-GM inequality, Lemma 2 and 4, we have

$$\Lambda_1 \leq \left(1 + \alpha_r^2\right)\mathbb{E}[\|(1-\alpha_r)(\nabla f(x^r) - g^r)\|^2] + 2\alpha_r L^2 U_r + 2\alpha_r^2\left(\frac{\sigma^2 + (L\rho)^2}{NK} + L^2 U_r\right).$$

Let $\sigma_\rho^2$ denote $\sigma^2 + (L\rho)^2$, which corresponds to the variance of the client's gradient after the perturbation is added. Using the AM-GM inequality again and Assumption 1, we have

$$\Lambda_1 \leq (1-\alpha_r)^2\left(1 + \alpha_r^2\right)\mathcal{E}_{r-1} + \left(1 + \frac{\alpha_r}{2}\right)L^2\mathbb{E}[\|x^r - x^{r-1}\|^2] + \frac{2\alpha_r^2\sigma_\rho^2}{NK} + 4\alpha_r L^2 U_r.$$

Finally,

$$\Lambda_1 \leq \left(1 - \frac{8\alpha_r}{9}\right)\mathcal{E}_{r-1} + 4\gamma^2\alpha_r L^2\mathbb{E}[\|\nabla f(x^{r-1})\|^2] + \frac{2\alpha_r^2\sigma_\rho^2}{NK} + 4\alpha_r L^2 U_r,$$

where we plug in $\|x^r - x^{r-1}\|^2 \leq 2\gamma^2(\|\nabla f(x^{r-1})\|^2 + \|g^r - \nabla f(x^{r-1})\|^2)$ and use $\gamma L \leq \frac{\alpha_r}{6}$ in the last inequality. Similarly for $r = 0$,

$$\Lambda_1 \leq \left(1 + \alpha_r^2\right)\mathbb{E}[\|(1-\alpha_r)(\nabla f(x^0) - g^0)\|^2] + 2\alpha_r L^2 U_0 + 2\alpha_r^2\left(\frac{\sigma_\rho^2}{NK} + L^2 U_0\right)$$

$$\leq (1-\alpha_r)\mathcal{E}_{-1} + \frac{2\alpha_r^2\sigma_\rho^2}{NK} + 4\alpha_r L^2 U_0.$$

Besides, by the AM-GM inequality and Lemma 1 and 4,

$$\Lambda_2 \leq \frac{1}{N}\sum_{i=1}^{N}\mathbb{E}\left[\left\|\frac{1}{K}\sum_k \nabla F(x_i^{r,k} + \delta_i^{r,k}; \xi_i^{r,k}) - c_i^r\right\|^2\right] \tag{6}$$

$$\leq 2K\sigma_\rho^2 + 2\sum_{i=1}^{N}\mathbb{E}\left[\left\|\frac{1}{K}\sum_k \nabla f_i(x_i^{r,k} + \delta_i^{r,k}) - c_i^r\right\|^2\right] \tag{7}$$

$$\leq 2K\sigma_\rho^2 + 6(L^2 U_r + L^2\mathbb{E}[\|x^r - x^{r-1}\|^2] + V_r). \tag{8}$$

Since $\mathbb{E}[\|x^r - x^{r-1}\|^2] \leq 2\gamma^2(\mathcal{E}_{r-1} + \mathbb{E}[\|\nabla f(x^{r-1})\|^2])$ and $\left(\frac{\alpha_r^2}{2} + 6\alpha_r^2 \frac{N-S}{S(N-1)}\right) 2(\gamma L)^2 \leq \frac{16\alpha_r}{9}(\gamma L)^2 \leq \frac{\alpha_r}{9}$,

we have

$$\mathcal{E}_r \leq \left(1 - \frac{8\alpha_r}{9}\right)\mathcal{E}_{r-1} + 16\alpha_r(\gamma L)^2 \mathbb{E}[\|\nabla f(x^{r-1})\|^2] + \frac{4\alpha_r^2 \sigma_\rho^2}{SK} + 10\alpha_r L^2 U_r + 6\alpha_r^2 \frac{N-S}{S(N-1)} V_r.$$

The case is similar for $r = 0$,

$$\mathcal{E}_0 \leq (1 - \alpha_r)\mathcal{E}_{-1} + \frac{2\alpha_r^2 \sigma_\rho^2}{NK} + 4\alpha_r L^2 U_0.$$

$\square$

**Lemma 6.** *If $\gamma L \leq \sqrt{\frac{1}{2\alpha_r}}$ and $\eta KL \leq \frac{1}{\alpha_r}$, it holds for all $r \geq 1$ that*

$$U_r \leq \eta^2 K^2 \left(8e(\mathcal{E}_{r-1} + 2\mathbb{E}[\|\nabla f(x^{r-1})\|^2] + \alpha_r^2 V_r) + \alpha_r^2 \sigma_\rho^2(K^{-1} + 2(\alpha_r \eta KL)^2)\right).$$

*Proof.* Recall that $\zeta_i^{r,k} = \mathbb{E}[x_i^{r,k+1} - x_i^{r,k} \mid \mathcal{F}_i^{r,k}] = -\eta(\alpha_r \nabla f_i(x_i^{r,k}) + (1 - \alpha_r)g^r - \alpha_r(c_i^r - c^r))$
and
$$\mathrm{Var}[x_i^{r,k+1} - x_i^{r,k} \mid \mathcal{F}_i^{r,k}] \leq \alpha_r^2 \eta^2 \sigma_\rho^2.$$

Then, from the definition of $U_r$ as the expected squared difference between each local model after $K$ local updates and the averaged model, we have:

$$U_r = \frac{1}{N} \sum_{i=1}^N \mathbb{E}\left[\|x_i^{r,K} - \bar{x}^{r,K}\|^2\right] \leq 2eK^2 \Xi_r + K\eta^2 \alpha_r^2 \sigma_\rho^2(1 + 2K^3 L^2 \eta^2 \alpha_r^2),$$

where the bound follows by analyzing the variance and bias accumulation over $K$ local steps.

Next, we estimate $\Xi_r$, the average squared norm of the update direction:

$$\Xi_r = \frac{\eta^2}{N} \sum_{i=1}^N \mathbb{E}[\|\alpha_r \nabla f_i(x^r) + (1 - \alpha_r)g^r - \alpha_r(c_i^r - c^r)\|^2]$$

$$= \frac{\eta^2}{N} \sum_{i=1}^N \mathbb{E}\Big[\big\|\alpha_r(\nabla f_i(x^r) - \nabla f_i(x^{r-1})) + (1 - \alpha_r)(g^r - \nabla f(x^{r-1}))$$

$$- \alpha_r(c_i^r - c^r - \nabla f_i(x^{r-1}) + \nabla f(x^{r-1})) + \nabla f(x^{r-1})\big\|^2\Big]$$

$$\leq 4\eta^2 \left(\alpha_r^2 L^2 \mathbb{E}[\|x^r - x^{r-1}\|^2] + (1 - \alpha_r)^2 \mathcal{E}_{r-1} + \alpha_r^2 V_r + \mathbb{E}[\|\nabla f(x^{r-1})\|^2]\right)$$

$$\leq 4\eta^2 \left(\mathcal{E}_{r-1} + 2\mathbb{E}[\|\nabla f(x^{r-1})\|^2] + \alpha_r^2 V_r\right).$$

Substituting the bound of $\Xi_r$ back into the expression for $U_r$, we obtain:

$$U_r \leq 2eK^2 \Xi_r + K\eta^2 \alpha_r^2 \sigma_\rho^2(1 + 2K^3 L^2 \eta^2 \alpha_r^2)$$

$$\leq 8e\eta^2 K^2(\mathcal{E}_{r-1} + 2\mathbb{E}[\|\nabla f(x^{r-1})\|^2] + \alpha_r^2 V_r) + \eta^2 K^2 \alpha_r^2 \sigma_\rho^2 \left(K^{-1} + 2(\alpha_r \eta KL)^2\right),$$

which proves the lemma. $\square$

**Lemma 7.** *Under the same conditions as Lemma 6, if $\alpha_r \eta KL \leq \frac{1}{24K^{1/4}}$ and $\eta K \leq \frac{5\gamma}{NS}$, then we have*

$$\sum_{r=0}^{R-1} V_r \leq \frac{3N}{S}\left(V_0 + \frac{4S}{NK}\sigma_\rho^2 + \frac{8N}{S}(\gamma L)^2 \sum_{r=-1}^{R-2}(\mathcal{E}_r + \mathbb{E}[\|\nabla f(x^r)\|^2])\right).$$

*Proof.* Since

$$c_i^{r+1} = \begin{cases} c_i^r, & \text{with probability } 1 - \frac{S}{N}, \\ \frac{1}{K}\sum_k \nabla F(x_i^{r,k} + \delta_i^{r,k}; \xi_i^{r,k}), & \text{with probability } \frac{S}{N}, \end{cases}$$

Using Young's inequality repeatedly, we have

$$
\begin{aligned}
V_{r+1} &= \left(1 - \frac{S}{N}\right) \frac{1}{N} \sum_{i=1}^{N} \mathbb{E}[\|c_i^r - \nabla f_i(x^r)\|^2] + \frac{S}{N} \frac{1}{N} \sum_{i=1}^{N} \mathbb{E}\left[\left\|\frac{1}{K} \sum_k \nabla F(x_i^{r,k} + \delta_i^{r,k}; \xi_i^{r,k}) - \nabla f_i(x^r)\right\|^2\right] \\
&\leq \left(1 - \frac{S}{N}\right) \frac{1}{N} \sum_{i=1}^{N} \mathbb{E}[\|c_i^r - \nabla f_i(x^r)\|^2] + \frac{S}{N}\left(2K\sigma_\rho^2 + 2L^2 U_r\right) \\
&\leq \left(1 - \frac{S}{N}\right) \frac{1}{N} \sum_{i=1}^{N} \mathbb{E}\left[\left(1 + \frac{S}{2N}\right)\|c_i^r - \nabla f_i(x^{r-1})\|^2 + \left(1 + \frac{2N}{S}\right) L^2 \|x^r - x^{r-1}\|^2\right] \\
&\quad + \frac{2S}{N}\left(\frac{\sigma_\rho^2}{K} + L^2 U_r\right) \\
&\leq \left(1 - \frac{2S}{N}\right) V_r + \frac{2N}{S} L^2 \mathbb{E}[\|x^r - x^{r-1}\|^2] + \frac{2N\sigma_\rho^2}{KS} + \frac{2S}{N} L^2 U_r.
\end{aligned}
$$

Here we apply Lemma 1 to obtain the second inequality. Combining this with Lemma6, we have

$$
\begin{aligned}
V_{r+1} &\leq \left(1 - \frac{2S}{N} + 16e\frac{S}{N}(\alpha_r \eta K L)^2\right) V_r + 2\sigma_\rho^2 \left(\frac{N}{KS} + \frac{2S}{N}(\alpha_r \eta K L)^2 (K - 1 + 2(\alpha_r \eta K L)^2)\right) \\
&\quad + \left(\frac{4S}{N}(\gamma L)^2 + \frac{32eS}{N}(\eta K L)^2\right)(\mathcal{E}_{r-1} + \mathbb{E}[\|\nabla f(x^{r-1})\|^2]).
\end{aligned}
$$

Finally,

$$
V_{r+1} \leq \left(1 - \frac{3S}{N}\right) V_r + \frac{4S}{NK}\sigma_\rho^2 + \frac{8S}{N}(\gamma L)^2(\mathcal{E}_{r-1} + \mathbb{E}[\|\nabla f(x^{r-1})\|^2]),
$$

where we apply the upper bound of $\eta$. Therefore, we finish the proof by summing up over $r$ from 0 to $R - 1$ and rearranging the inequality. $\qquad\square$

**Theorem 1.** *Under Assumption 1 and 2, if we take $g^0 = 0$, $c_i^0 = \frac{1}{B} \sum_{b=1}^{B} \nabla F(x^0; \xi_i^b)$ with $\{\xi_i^b\}_{b=1}^{B} \overset{iid}{\sim} D_i$, $c^0 = \frac{1}{N} \sum_{i=1}^{N} c_i^0$, and set*

$$
\gamma = \frac{\alpha_r}{L}, \quad \alpha_r = \min\left\{c, \frac{S}{N^{2/3}}, \frac{\sqrt{L\Delta SK}}{\sigma_\rho^2 R}, \frac{\sqrt{L\Delta S^2}}{G_0 N}\right\},
$$

$$
\eta K L \lesssim \min\left\{\frac{1}{\sqrt{S}}, \frac{1}{\alpha_r K^{1/4}}, \frac{\sqrt{S}}{N}\right\}, \quad B = \left\lceil \frac{NK}{SR} \right\rceil,
$$

*then FedWMSAM converges as*

$$
\frac{1}{R} \sum_{r=0}^{R-1} \mathbb{E}[\|\nabla f(x^r)\|^2] \lesssim \sqrt{\frac{L\Delta \sigma_\rho^2}{SKR}} + \frac{L\Delta}{R}\left(1 + \frac{N^{2/3}}{S}\right).
$$

*Proof.* By Lemma 5, summing over $r$ from 0 to $R-1$ and plugging Lemma 6 and 7 in, we have

$$\sum_{r=0}^{R-1} \mathcal{E}_r \leq \left(1 - \frac{8\alpha_r}{9}\right) \sum_{r=-1}^{R-2} \mathcal{E}_r + 16\alpha_r(\gamma L)^2 \sum_{r=0}^{R-2} \mathbb{E}[\|\nabla f(x^r)\|^2]$$

$$+ \frac{4\alpha_r^2\sigma_\rho^2}{SK}R + 10\alpha_r L^2 \sum_{r=0}^{R-1} U_r + 6\alpha_r^2 \frac{N-S}{S(N-1)} \sum_{r=0}^{R-1} V_r$$

$$\leq \left(1 - \frac{8\alpha_r}{9} + 80e\alpha_r(\eta KL)^2\right) \sum_{r=-1}^{R-2} \mathcal{E}_r + \left(16\alpha_r(\gamma L)^2 + 160e\alpha_r(\eta KL)^2\right) \sum_{r=0}^{R-2} \mathbb{E}[\|\nabla f(x^r)\|^2]$$

$$+ \alpha_r^2\sigma_\rho^2 R\left(\frac{4}{SK} + 10(\eta KL)^2(K - 1 + 2(\alpha_r\eta KL)^2)\right)$$

$$+ \alpha_r^2 \left(6\frac{N-S}{S(N-1)} + 80e\alpha_r(\eta KL)^2\right) \sum_{r=0}^{R-1} V_r$$

$$\leq \left(1 - \frac{7\alpha_r}{9}\right) \sum_{r=-1}^{R-2} \mathcal{E}_r + \left(16\alpha_r(\gamma L)^2 + \frac{\alpha_r}{9}\right) \sum_{r=0}^{R-2} \mathbb{E}[\|\nabla f(x^r)\|^2]$$

$$+ \frac{80\alpha_r^2\sigma_\rho^2}{SK}R + \frac{30\alpha_r^2 N}{S^2}V_0.$$

Here the coefficients in the last inequality are derived by the following bounds:

$$\begin{cases} 160e\alpha_r(\eta KL)^2 + 24\left(\frac{\alpha_r\gamma LN}{S}\right)^2 \left(\frac{6(N-S)}{S(N-1)} + 80e\alpha_r(\eta KL)^2\right) \leq \frac{\alpha_r}{9}, \\ 10(\eta KL)^2(K - 1 + 2(\alpha_r\eta KL)^2) + 960e\alpha_r K^{-1}(\eta KL)^2 \leq \frac{4}{SK}, \\ 80e\alpha_r(\eta KL)^2 \leq \frac{S}{4}. \end{cases}$$

which can be guaranteed by

$$\begin{cases} \gamma L \lesssim \frac{S^{3/2}}{\alpha_r^{1/2}N}, \\ \eta KL \lesssim \frac{1}{S^{1/2}}. \end{cases}$$

Therefore,

$$\sum_{r=0}^{R-1} \mathcal{E}_r \leq \frac{7\alpha_r}{9}\mathcal{E}_{-1} + \frac{2}{7} \sum_{r=-1}^{R-2} \mathbb{E}[\|\nabla f(x^r)\|^2] + \frac{270\alpha_r N}{7S^2}V_0 + \frac{720\alpha_r\sigma_\rho^2}{7SK}R.$$

Combining this inequality with Lemma 1, we obtain

$$\frac{\gamma}{L}\mathbb{E}[f(x^R) - f(x^0)] \leq -\frac{1}{7} \sum_{r=0}^{R-1} \mathbb{E}[\|\nabla f(x^r)\|^2] + \frac{39\alpha_r}{56}\mathcal{E}_{-1} + \frac{585\alpha_r N}{28S^2}V_0 + \frac{390\alpha_r\sigma_\rho^2}{7SK}R.$$

Finally, noticing that $g^0 = 0$ implies $\mathcal{E}_{-1} \le 2L\Delta$ and $c_i = \frac{1}{B}\sum_b \nabla F(x^0; \xi_i^b)$ implies $V_0 \le \frac{\sigma_\rho^2}{B} \le \frac{\sigma_\rho^2 SR}{NK}$, we reach

$$\frac{1}{R}\sum_{r=0}^{R-1}\mathbb{E}[\|\nabla f(x^r)\|^2] \lesssim \frac{L\Delta}{\gamma LR} + \frac{\mathcal{E}_{-1}}{\alpha_r R} + \frac{\alpha_r N}{S^2 R}V_0 + \frac{\alpha_r \sigma_\rho^2}{SK}$$

$$\lesssim \frac{L\Delta}{\alpha_r R} + \frac{L\Delta}{S^{3/2}R}N^{1/2} + \frac{\alpha_r \sigma_\rho^2}{SK}$$

$$\lesssim \frac{L\Delta}{R}\left(1 + \frac{N^{2/3}}{S}\right) + \sqrt{\frac{L\Delta\sigma_\rho^2}{SKR}}.$$

$\square$

### A.2 Comparison of Convergence Rates

Table 7 summarizes the theoretical convergence rates of FedWMSAM and several representative federated optimization algorithms under non-convex settings. These methods differ in their design focuses—some emphasize global convergence guarantees, while others incorporate mechanisms for bias correction or perturbation modeling.

Table 7: Theoretical convergence rates of FedWMSAM and related federated optimization algorithms under non-convex settings.

| Method | Convergence Rate (in expectation) |
|---|---|
| **FedWMSAM (Ours)** | $\frac{1}{R}\sum_{r=0}^{R-1}\mathbb{E}[\|\nabla f(x^r)\|^2] \lesssim \sqrt{\frac{L\Delta\sigma_\rho^2}{SKR}} + \frac{L\Delta}{R}\left(1 + \frac{N^{2/3}}{S}\right)$ |
| FedAvg [1] | $\frac{1}{R}\sum_{r=0}^{R-1}\mathbb{E}[\|\nabla f(x^r)\|^2] \le \mathcal{O}\left(\frac{1}{\sqrt{KR}}\right) + \text{bias terms}$ |
| Scaffold [11] | $\frac{1}{R}\sum_{r=0}^{R-1}\mathbb{E}[\|\nabla f(x^r)\|^2] \le \mathcal{O}\left(\frac{1}{KR} + \frac{1}{R} + \frac{1}{K}\right)$ |
| FedCM [22] | $\mathbb{E}[\|\nabla f(x^r)\|^2] \le \mathcal{O}\left(\frac{1}{\sqrt{KR}}\right) + \text{momentum terms}$ |
| FedSAM [26] | $\frac{1}{T}\sum_{t=1}^{T}\mathbb{E}[\|\nabla f(w_t)\|^2] \le \mathcal{O}\left(\frac{1}{\sqrt{TEN}}\right) + \text{perturbation terms}$ |
| MoFedSAM [26] | $\frac{1}{T}\sum_{t=1}^{T}\mathbb{E}[\|\nabla f(w_t)\|^2] \le \mathcal{O}\left(\frac{1}{\sqrt{TEN}}\right) + \text{mom.\&per. terms}$ |
| FedLESAM [36] | $\frac{1}{T}\sum_{t=1}^{T}\mathbb{E}[\|\nabla f(w_t)\|^2] \le \mathcal{O}\left(\frac{1}{\sqrt{TEN}}\right) + \text{perturbation terms}$ |
| FedSMOO [28] | $\frac{1}{T}\sum_{t=1}^{T}\mathbb{E}[\|\nabla f(w_t)\|^2] \le \frac{1}{\zeta\beta T}\left(\kappa_f + \frac{3\beta L\kappa_r}{n} + \frac{72\beta^2 L^2 m\delta_0}{n}\right)$ |

Our proposed FedWMSAM achieves the following convergence bound:

$$\frac{1}{R}\sum_{r=0}^{R-1}\mathbb{E}[\|\nabla f(x^r)\|^2] \lesssim \sqrt{\frac{L\Delta\sigma_\rho^2}{SKR}} + \frac{L\Delta}{R}\left(1 + \frac{N^{2/3}}{S}\right),$$

where the first term captures the influence of gradient noise introduced by sharpness-aware perturbations, and the second term reflects the effects of client sampling and system heterogeneity.

This result highlights three key aspects of FedWMSAM:

1. It explicitly models the impact of perturbation-induced gradient variance $\sigma_\rho^2$, improving robustness in sharp-loss regions;

2. It incorporates the effects of core federated parameters, including local steps $K$, communication rounds $R$, sampled clients $S$, and total data size $N$;

3. It employs a momentum-driven adaptive reweighting strategy that enhances convergence stability, particularly under non-IID and long-tailed client distributions.

Under standard assumptions (e.g., smoothness and bounded variance), the convergence rate of FedWMSAM is comparable to or potentially sharper than that of FedAvg, Scaffold, and FedSAM-like methods. Importantly, it achieves a favorable trade-off between variance reduction and bias mitigation, which is especially beneficial in realistic federated learning scenarios where client data distributions are heterogeneous and participation is imbalanced.

Table 8: Comparison of additional theoretical properties across methods.

| Method | Non-IID Support | Bias Control | Gradient Noise Aware |
|---|---|---|---|
| FedAvg | No | No | No |
| FedCM | No | No | No |
| Scaffold | Yes | Yes | No |
| FedSAM | Partial | No | Yes |
| MoFedSAM | Partial | No | Yes |
| FedLESAM | Yes | Yes | Yes |
| FedSMOO | Yes | Yes | Yes |
| **FedWMSAM (Ours)** | Yes | Yes | Yes |

Table 8 complements the convergence rate analysis by qualitatively comparing theoretical attributes of the listed methods. "Non-IID Support" refers to whether a method explicitly addresses statistical heterogeneity across clients. "Bias Control" denotes the ability to mitigate client drift or sampling bias, often via correction mechanisms. "Gradient Noise Aware" indicates whether a method takes into account the influence of stochastic or perturbation-induced noise in gradient estimation.

FedWMSAM stands out by simultaneously addressing all three aspects. In contrast, methods like FedAvg and FedCM offer simplicity but lack explicit mechanisms for bias correction or heterogeneity handling. Scaffold effectively controls client drift through control variates but does not incorporate perturbation modeling. FedSAM-like methods consider gradient noise via sharpness-aware perturbations, but their support for non-IID settings is indirect and lacks dedicated bias correction.

Overall, FedWMSAM provides a unified approach that combines variance reduction, non-IID robustness, and perturbation modeling, resulting in a theoretically grounded and practically effective convergence guarantee for challenging federated learning environments.

# B  Ablation Study

## B.1  Rationale Behind Using Cosine Similarity for $\alpha_r$ Calculation: Experimental Validation

To evaluate the appropriateness of using the Mean Cosine Similarity between the Local Gradient and Momentum as an indicator of the training phase, we conducted an experiment to analyze its correlation with test accuracy trends over time.

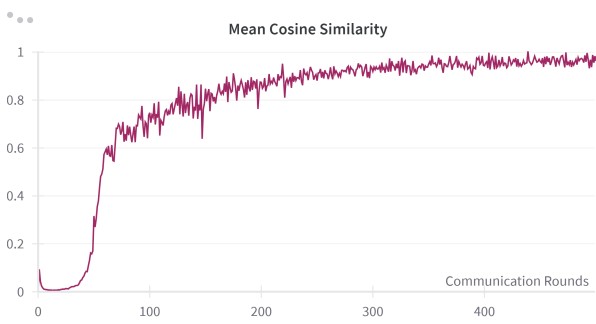

Figure 7: Mean Cosine Similarity between Local Gradient and Momentum

The training process typically enters its later stages when the test accuracy stabilizes. We examined the relationship between the cosine similarity and accuracy by plotting their respective trends. As shown in Figure 7, the cosine similarity increases rapidly around the 50th round and reaches a steady state after around 200 rounds. This aligns with the accuracy curve shown in Figure 8, where accuracy also rises quickly before 50 rounds, decelerates between 50 and 200 rounds, and then stabilizes after 200 rounds.

From this analysis, we observe a negative correlation between the Mean Cosine Similarity and the speed of accuracy improvement. Specifically, the slope of the accuracy curve appears to be approximately one minus the cosine similarity.

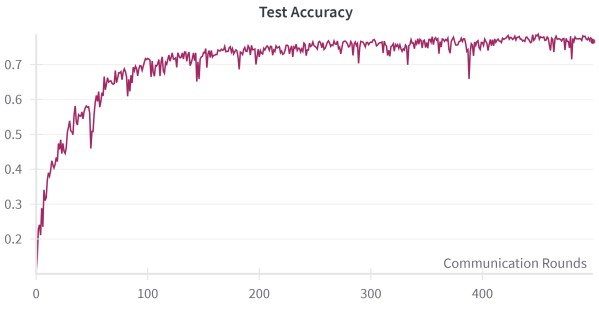

Figure 8: Test Accuracy

To further explore this relationship, we plotted one minus the cosine similarity value in Figure 9. The plot clearly demonstrates that a higher cosine similarity corresponds to a more rapid increase in test accuracy, while lower similarity values indicate slower accuracy growth.

Based on these findings, we established a clear connection between test accuracy and cosine similarity, which can be used to estimate the training period. As discussed previously, the momentum should have a more substantial influence in the earlier stages of training and diminish in importance later. Therefore, we chose the Mean Cosine Similarity as the basis for calculating the weighting factor $\alpha_r$, thereby enabling us to dynamically adjust the role of momentum during training.

## B.2  Impact of Decay Coefficient $\lambda$ on $\alpha_r$ Update and Performance Optimization

The decay factor $\lambda$ plays a pivotal role in controlling the rate of change for $\alpha_r$ during the update process:

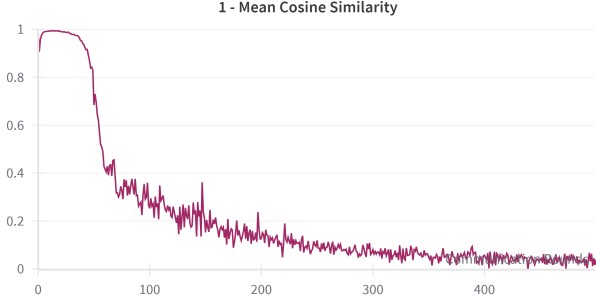

Figure 9: One Minus Mean Cosine Similarity

$$\alpha_{r+1} = (1 - \lambda)\,\alpha_r + \lambda\,\min\Big(\max\big(\hat{\alpha}_{r+1},\,0.1\big),\,0.9\Big)$$

This update equation allows for a smooth adjustment of $\alpha_r$, with $\lambda$ determining how quickly $\alpha_r$ adapts to new values. However, as observed, the Mean Cosine Similarity shows significant fluctuations, which can negatively affect the model's training. Therefore, selecting an optimal value for $\lambda$ is crucial for ensuring stable and effective training.

To assess the impact of $\lambda$, we trained the model with different values of $\lambda$ and analyzed both accuracy and changes in $\alpha_r$.

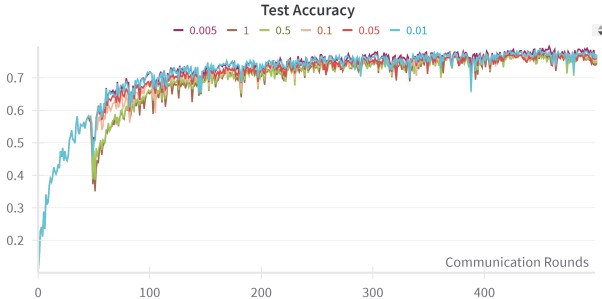

Figure 10: Test Accuracy for Different $\lambda$ Values on CIFAR-10 under Dirichlet $\beta = 0.1$

As shown in Figure 10, the setting $\lambda = 0.01$ consistently outperforms other values throughout training, achieving higher overall accuracy.

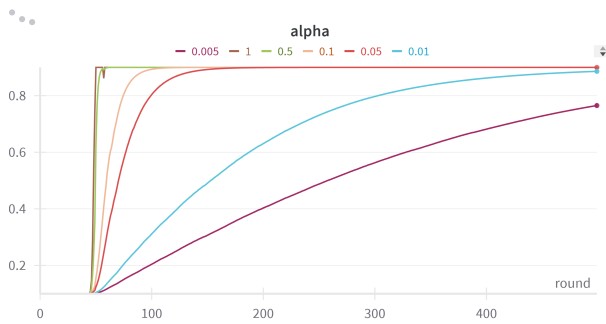

Figure 11: Variation of $\alpha_r$ Across Training Rounds for Different $\lambda$ Values on CIFAR-10 under Dirichlet $\beta = 0.1$

Additionally, we observed the evolution of $\alpha_r$ over the rounds, which is shown in Figure 11. This figure shows how different $\lambda$ values affect the rate of change in $\alpha_r$, demonstrating that $\lambda$ significantly influences the training dynamics.

We further evaluated the performance by calculating the number of rounds required to reach various accuracy levels for each $\lambda$ value. As shown in Table 9, the value $\lambda = 0.01$ consistently required fewer rounds to reach higher accuracy levels, making it the most efficient choice in most cases.

Table 9: Rounds to Achieve Different Accuracy Levels for Various $\lambda$ Values

| $\lambda$ | 0.68 | 0.7 | 0.72 | 0.74 | 0.76 | 0.78 |
|---|---|---|---|---|---|---|
| 1 | 119 | 144 | 170 | 224 | 343 | - |
| 0.5 | 118 | 152 | 170 | 241 | 313 | - |
| 0.1 | 99 | 123 | 153 | 208 | 311 | - |
| 0.05 | 97 | 113 | 153 | 207 | 292 | - |
| 0.01 | 72 | 98 | 114 | 153 | 241 | 356 |
| 0.005 | 72 | 98 | 114 | 153 | 234 | 382 |

In summary, the choice of decay factor $\lambda$ is crucial for balancing model stability and training speed. Based on our findings, $\lambda = 0.01$ strikes the best balance, yielding faster convergence and improved model performance.

### B.3   Limitation Analysis

**Dataset-Specific Sensitivity of $\lambda$.** The choice of the decay factor $\lambda$, which governs the update dynamics of $\alpha_r$, plays a critical role in training stability and performance. However, our findings suggest that the optimal value of $\lambda$ may vary across different datasets and distribution settings. This sensitivity could pose challenges for practitioners seeking to apply the method in diverse real-world scenarios, as it introduces an additional layer of hyperparameter tuning.

**Instability Caused by Cosine Similarity Fluctuations.** The use of Mean Cosine Similarity as an indicator of training-phase progression enables adaptive momentum adjustment. Nevertheless, this metric exhibits significant short-term fluctuations, which may lead to unstable or overly aggressive updates in $\alpha_r$, particularly in early training rounds. While we mitigate this using a smoothing mechanism via $\lambda$, the inherent volatility of cosine similarity remains a potential source of instability.

**Limited Scope of Experimental Evaluation.** Our experimental validation primarily focuses on image classification tasks under IID and non-IID settings. While the proposed method demonstrates strong performance in these benchmarks, its applicability to other types of federated learning problems—such as natural language processing or graph learning—remains unexplored. Broader validation across tasks and modalities would strengthen the generalizability of our approach.

**Lack of Theoretical Justification for Momentum-Guided SAM.** Although our work introduces a novel use of momentum to guide the SAM update direction and empirically validates its effectiveness, we fall short of providing a rigorous theoretical analysis to substantiate the superiority of this strategy. The current theoretical contribution is limited to convergence guarantees, without a formal justification of how momentum-driven perturbations improve generalization or optimization efficiency in federated settings.

## C  Visualization Results

### C.1  t-SNE Visualization Comparison

To gain deeper insight into the representations learned by different methods, we visualize the feature embeddings of the global models using t-SNE [44]. Specifically, we compare FedAvg [1], FedCM [22], Scaffold [11], FedSAM [26], MoFedSAM [26],FedLESAM [36], FedSMOO [28] and our proposed FedWMSAM across different training stages.

Figure 12 presents the t-SNE plots of model embeddings after 200 communication rounds on the CIFAR-10 dataset using ResNet-18. Compared to other baselines, FedWMSAM yields more clearly separated and compact clusters. This suggests that our method encourages more discriminative feature representations and facilitates better client-level generalization, even in early training stages.

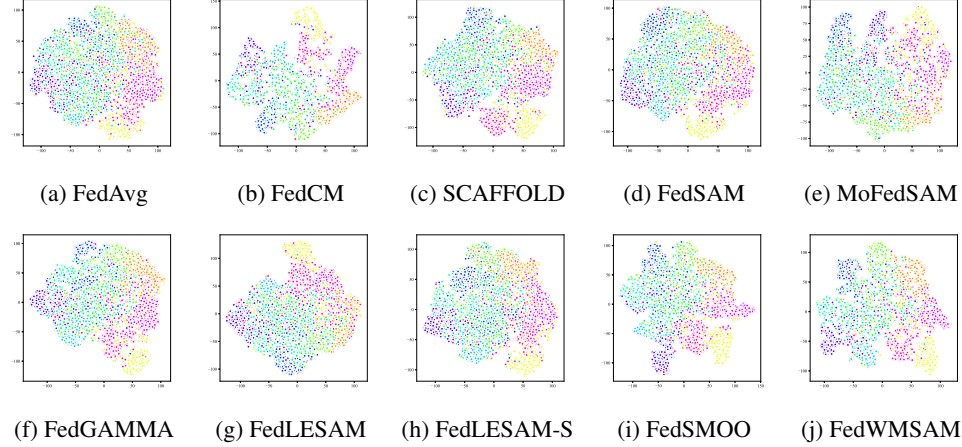

(a) FedAvg     (b) FedCM     (c) SCAFFOLD     (d) FedSAM     (e) MoFedSAM

(f) FedGAMMA     (g) FedLESAM     (h) FedLESAM-S     (i) FedSMOO     (j) FedWMSAM

Figure 12: t-SNE visualization of global model embeddings after 200 communication rounds.

To further assess the quality and stability of the learned representations at convergence, we visualize the embeddings after 1000 communication rounds in Figure 13. The clusters produced by FedWMSAM remain well-separated and exhibit relatively uniform spatial distribution, indicating enhanced inter-class separability and flatter decision boundaries. We attribute this behavior to the momentum-driven adaptive reweighting and perturbation-aware updates, which help guide the optimization toward flatter, more generalizable minima. It is worth noting that although FedSMOO [28] achieves comparable cluster separation, this advantage comes at over twice the computational and communication overhead as our method.

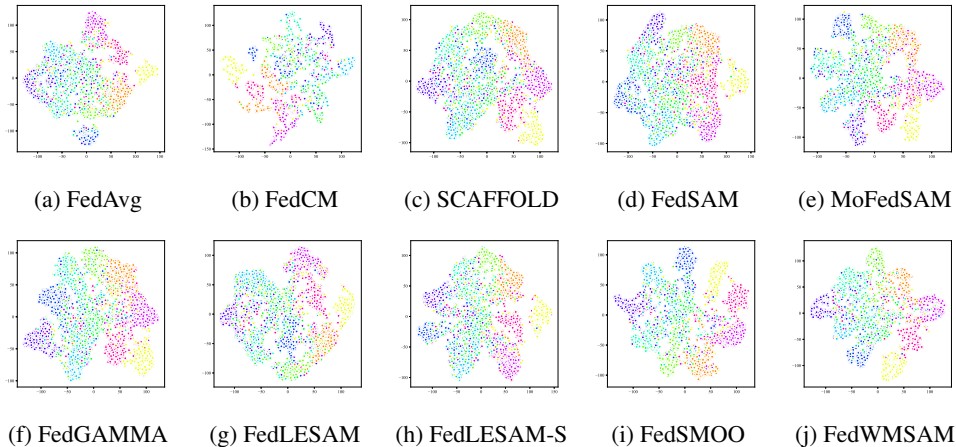

(a) FedAvg     (b) FedCM     (c) SCAFFOLD     (d) FedSAM     (e) MoFedSAM

(f) FedGAMMA     (g) FedLESAM     (h) FedLESAM-S     (i) FedSMOO     (j) FedWMSAM

Figure 13: t-SNE visualization of global model embeddings after 1000 communication rounds.

## C.2 Performance Analysis and Comparative Evaluation

In this section, we present a detailed analysis of the experimental results, accompanied by visualizations to compare the performance of our proposed FedWMSAM method against several state-of-the-art approaches.

We begin by presenting the results on CIFAR-10, as shown in Figures 14 and 15. The blue and green lines represent the FedWMSAM method, which demonstrates superior performance, with faster convergence and higher accuracy than other methods. The experiments were conducted with Dirichlet distribution parameters $\beta = 0.1$ and pathological heterogeneity $\gamma = 3$.

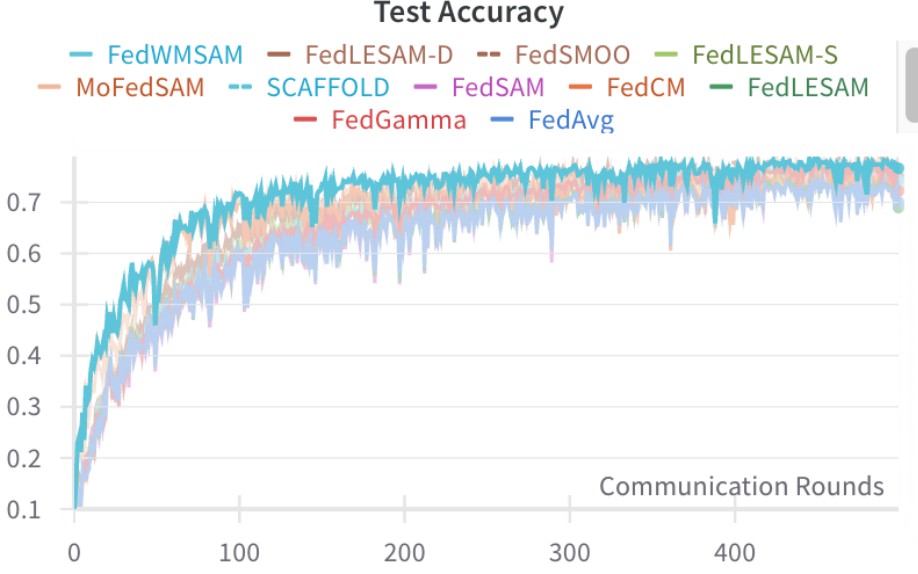

Figure 14: Test Accuracy on CIFAR-10 under Dirichlet $\beta = 0.1$

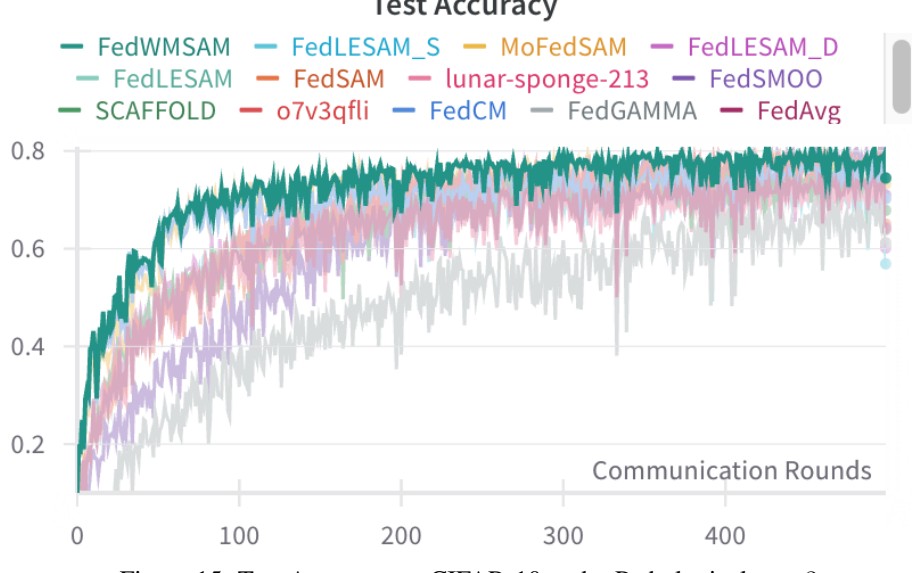

Figure 15: Test Accuracy on CIFAR-10 under Pathological $\gamma = 3$

Next, Figures 16 and 17 illustrate the results on CIFAR-100. The orange, gray, and pink lines represent the FedWMSAM method under different experimental conditions. In these experiments, the data were split using Dirichlet $\beta = 0.1$ and pathological $\gamma = 10$. Notably, under extreme heterogeneity with $\gamma = 10$ (as shown in Figure 17), the gray line representing our method clearly outperforms all other methods in terms of both convergence speed and accuracy, demonstrating the robustness of FedWMSAM in challenging scenarios.

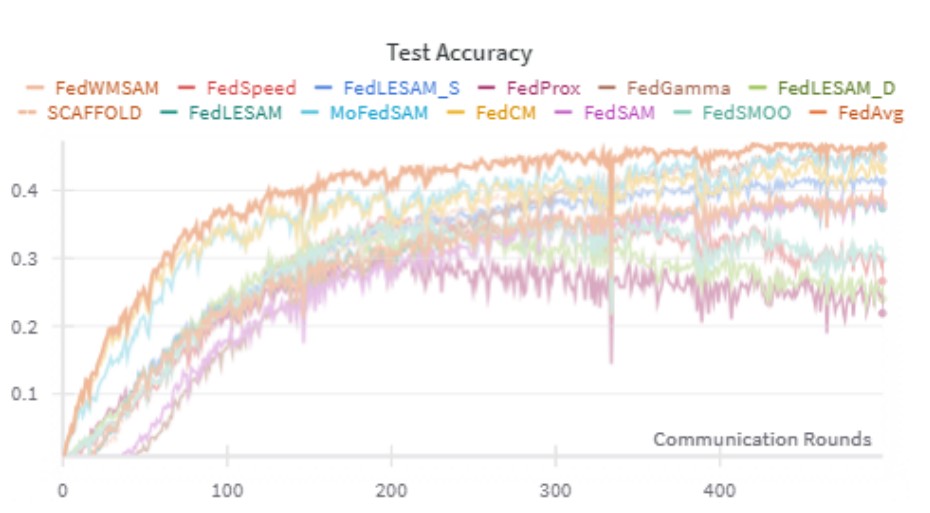

Figure 16: Test Accuracy on CIFAR-100 under Dirichlet $\beta = 0.1$

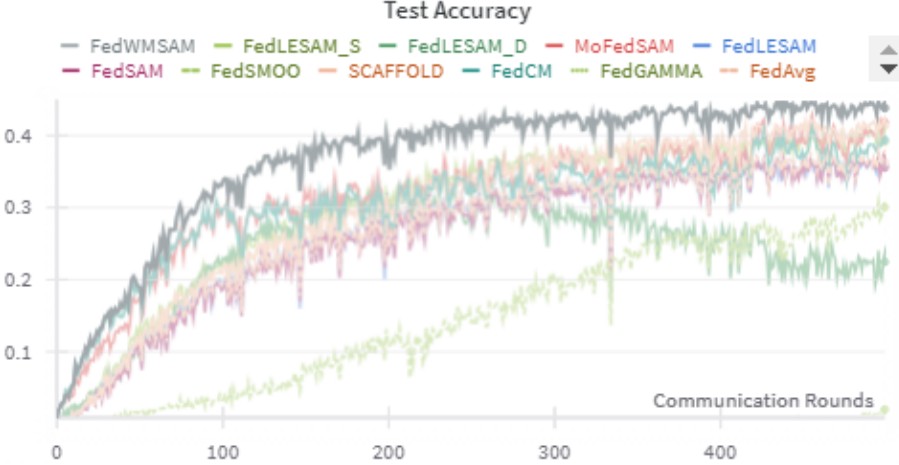

Figure 17: Test Accuracy on CIFAR-100 under Pathological $\gamma = 10$

This analysis highlights the effectiveness of FedWMSAM, particularly under heterogeneous conditions, and demonstrates its ability to achieve faster convergence and superior accuracy compared to alternative methods across a range of datasets.

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
