# OpenReview forum: "FedWMSAM: Fast and Flat Federated Learning via Weighted Momentum and Sharpness-Aware Minimization"
_NeurIPS.cc/2025/Conference — NeurIPS 2025 poster_

### Official Review · Reviewer_3mtU · 2025-06-04

**Clarity:** 3
**Significance:** 2
**Originality:** 3
**Rating:** 4
**Confidence:** 4

**Summary:**

FL in edge computing faces several unique challenges, including limited communication rounds, data heterogeneity, and client drift, which lead to slow convergence and poor model generalization. To seamlessly concatenate Momentum and SAM while avoiding their flaws, this paper proposes a novel approach called FedWMSAM to achieve fast and flat FL. Then it estimates global flat minima and uses a dynamic weighting mechanism to balance the contributions of momentum and SAM adaptively throughout the training process. Extensive experiments validate its motivation.

**Questions:**

1. **Experiments:** From the experimental results in Table 1, we can see that when the heterogeneity of the data set is weak, the effect of the method proposed in this paper is not outstanding. What is the reason for this? How can this problem be improved?
2. **Theoretical analysis:** In theoretical analysis, the dynamic adjustment weight is directly represented by a variable. Can its dynamic design be reflected or discussed in theoretical analysis?

**Ethical Concerns:**

["NO or VERY MINOR ethics concerns only"]

**Final Justification:**

The authors have improved the rigor of the experiments and the problem statement in this work. The argumentation has also been improved. Considering the interest of the questions raised and the completeness of the theoretical proof, I have increased the score by one.

**Limitations:**

1. **Motivation and novelty:** This motivation (like limited communication rounds, data heterogeneity, and client drift and slow convergence and poor model generalization) is quite commonly seen in FL. This writing style is highly repetitive with other similar works and does not highlight one's own insights, especially in the abstract and introduction.
2. **Contribution:** The core technology of this work seems to be a combination of the momentum and the SAM with an adaptive weighted scheme. What are the key technical contributions or the technical novelty?

**Quality:**

3

**Strengths And Weaknesses:**

**Strengths:**
1. **Clear motivation and problem formulation:** This paper clearly identifies the core challenges in federated learning (limited communication, data heterogeneity, and client drift) and links them directly to the need for faster convergence and better generalization, which momentum and SAM individually address.
2. **Extensive experimental validation:** This paper claims extensive experiments on multiple datasets and model architectures, which is promising for demonstrating the method's effectiveness, adaptability, and robustness.
3. **Good writing and framework:** This paper is well-organized and easy to read.

Weakness:
See questions and limitations below.

---

> ### Author Rebuttal · Authors · 2025-07-31
>
> ### Q1: Experiments
> **Question**: From the experimental results in Table 1, we can see that when the heterogeneity of the data set is weak, the effect of the method proposed in this paper is not outstanding. What is the reason for this? How can this problem be improved?
>
> **Ans1**: Thank you for your attention to the experimental results in Table 1. Our method performs less prominently in scenarios with weak data heterogeneity mainly because traditional methods (e.g., FedAvg) experience less performance degradation under such conditions, and the differences between clients have a smaller impact on the global model, making it harder for the momentum mechanism and adaptive weighting to demonstrate their advantages. However, in scenarios with high heterogeneity, these improvements significantly enhance performance. Additionally, compared to other methods (e.g., FedSMOO), our method is slightly less competitive in low heterogeneity settings, but those methods generally incur higher computational and communication costs (approximately twice as much as ours), while our approach is specifically designed for resource-constrained edge environments.
>
> ---
>
> ### Q2: Theoretical Analysis
> **Question**: In theoretical analysis, the dynamic adjustment weight is directly represented by a variable. Can its dynamic design be reflected or discussed in theoretical analysis?
>
> **Ans2**: We appreciate the reviewer’s attention to the theoretical analysis of dynamic weight adjustment. Dynamic weight adjustment significantly improves model performance in practice, primarily by balancing the contributions of the momentum mechanism and the Sharpness-Aware Minimization (SAM) method at different stages. However, in the theoretical analysis, dynamic weight adjustment does not directly affect the convergence properties. Its main role lies in optimizing the flatness of the global model through weight adjustment rather than directly influencing the convergence rate.
>
> Additionally, we have provided a detailed analysis in Appendix B on the impact and selection of dynamic weights in experiments. The experimental results further validate its effectiveness, complementing the limitations in the theoretical analysis.
>
> ---
>
> ### Q3: Motivation and Novelty
> **Question**: This motivation (like limited communication rounds, data heterogeneity, and client drift and slow convergence and poor model generalization) is quite commonly seen in FL. This writing style is highly repetitive with other similar works and does not highlight one's own insights, especially in the abstract and introduction.
>
> **Ans3**: We appreciate the reviewer’s suggestions regarding the abstract and introduction. We recognize that these sections in the original version may not have sufficiently highlighted the core problem and innovations of this work, which might have caused some overlap with other similar studies. In fact, the core focus of our work is to address the global optimization challenges in federated learning caused by data heterogeneity, particularly the limitations of SAM in this context. By introducing a momentum mechanism and adaptive weighting strategy, we significantly improve the application of SAM, achieving better performance in heterogeneous data scenarios.
>
> In the camera-ready version, we will refine the abstract and introduction to emphasize our key innovation: how solving for globally flat optima at the local level helps overcome the challenges posed by heterogeneity. This will further highlight the uniqueness and contributions of our work.
>
> ---
>
> ### Q4: Contribution
> **Question**: The core technology of this work seems to be a combination of the momentum and the SAM with an adaptive weighted scheme. What are the key technical contributions or the technical novelty?
>
> **Ans4**: We appreciate the reviewer’s attention to the technical contributions of our work. Our core contributions go beyond the simple combination of momentum and SAM and include the following key innovations:
>
> 1. **Systematic Analysis of the Core Contradiction in Applying SAM to Federated Learning**:
>    We are the first to systematically analyze the fundamental contradiction of applying SAM in federated learning: SAM relies on perturbation computation based on local client data, while improving the generalization performance of the global model requires optimizing SAM at the global level. This contradiction is a critical challenge in federated learning, yet it has been overlooked in prior studies. To address this issue, we propose leveraging a momentum mechanism to transmit global information across clients with low communication costs, effectively guiding local optimization and mitigating this contradiction.
>
> 2. **Proposal of the FedWMSAM Method and Convergence Proof**:
>    We propose **FedWMSAM**, a novel method that effectively combines personalized momentum with SAM to address the divergence between local and global models. FedWMSAM introduces an adaptive adjustment factor that dynamically balances the contributions of momentum and perturbation at different training stages. This ensures efficient convergence to a flatter global optimum under heterogeneous conditions. Additionally, we theoretically prove the convergence of FedWMSAM, demonstrating its efficiency with a convergence rate of **O(LΔσ²/(ρNKR) + LΔ/R)**.
>
> 3. **Comprehensive Experimental Validation**:
>    We conducted extensive experiments across **4 image datasets (Fashion-MNIST, CIFAR-10, CIFAR-100, OfficeHome)**, **9 state-of-the-art methods**, and various heterogeneous settings. The results demonstrate that FedWMSAM significantly outperforms existing methods in terms of accuracy, convergence speed, and computational cost. Ablation and sensitivity analyses further validate the robustness of our method:
>    - **Accuracy**: FedWMSAM achieves consistent accuracy improvements across datasets.
>    - **Convergence Speed**: It converges faster than existing methods, even under high heterogeneity.
>    - **Computational Efficiency**: FedWMSAM reduces computational costs while achieving superior performance.
>
> 4. **Theoretical and Practical Insights**:
>    FedWMSAM extends the application of SAM to federated learning by resolving the local-global optimization contradiction. The theoretical convergence proof not only establishes its efficiency but also provides a foundation for future studies to address similar challenges in federated optimization.
>
> 5. **Comparison with Existing Work and Improvements**:
>    Our method builds on ideas from related works (e.g., FedAvg, FedSAM, and FedLESAM) but introduces significant improvements. Unlike existing methods, FedWMSAM effectively addresses the heterogeneity-induced divergence between local and global models and achieves better performance with lower computational and communication costs.
>
> In the **camera-ready version**, we will refine the contribution summary to ensure clarity and emphasize the following key points:
> - The identification of the core contradiction in applying SAM to federated learning and how FedWMSAM resolves it.
> - The innovative combination of momentum and SAM, supported by adaptive weighting, to dynamically balance local and global optimization.
> - The theoretical convergence proof and its practical significance in federated learning scenarios.
> - Extensive experimental results that establish FedWMSAM’s superiority across diverse settings.
>
> Thank you again for your valuable feedback, which has helped us better articulate the novelty and significance of our work.

---

> ### Comment · Reviewer_3mtU · 2025-08-04
> **Reply to the authors.**
>
> Q1: I think the authors should provide additional experimental comparisons with dir0.3 to better clarify the boundaries of heterogeneous selection. Due to time constraints, the authors may wish to showcase the best algorithms.
>
> Q3: I understand the authors' intentions. The question is very intuitive and interesting, and the writing needs to be more precise.
>
> Therefore, I decided to maintain my score.

---

> ### Author Response · Authors · 2025-08-05
> **Thank You for Your Constructive Feedback and Suggestions**
>
> Thank you for your thoughtful and constructive feedback. We greatly value your insightful comments, and they have been immensely helpful in guiding us to improve our work. Below, we address your comments in detail:
>
> ### **Q1: Additional experimental comparisons with dir0.3 to clarify boundaries of heterogeneous selection.**
>
> Thank you for pointing out the importance of evaluating our method under Dirichlet 0.3 to better characterize the boundaries of heterogeneous selection. We appreciate this valuable suggestion and have conducted the requested experiments accordingly.
>
> In our original submission, we adopted Dirichlet 0.1 and 0.6 following common practices in prior work (e.g., FedCM[1], FedSMOO[2], FedLESAM[3]) to capture both extreme and moderate heterogeneity scenarios. These choices were not to cherry-pick favorable settings, but to ensure broad coverage and fair comparisons.
>
> In response to your comment, we have conducted new experiments on **CIFAR-10 under Dirichlet 0.3**, and the results are as follows:
>
> - **FedSMOO**: 79.52%
> - **FedWMSAM (Ours)**: **80.12%**
>
> These results confirm the robustness of FedWMSAM even in intermediate heterogeneity levels. We are currently expanding this analysis to include **Fashion-MNIST**, **CIFAR-10**, and **CIFAR-100** under Dirichlet 0.3, and the complete results will be included in the revised version.
>
> As for your observation that **FedSMOO performs slightly better than our method on Fashion-MNIST and CIFAR-10**, we would like to clarify the following:
>
> - These datasets are relatively simple, and FedSMOO achieves its performance through significantly **heavier computation and regularization**. Specifically, FedSMOO requires **29.77 seconds per round** in line 292, nearly **twice the computation time** of our method (**15.03 seconds per round**).
> - Our design prioritizes both performance and efficiency. On more complex benchmarks such as **CIFAR-100** and **OfficeHome**, which better reflect real-world heterogeneity, **FedWMSAM consistently outperforms FedSMOO**, demonstrating superior scalability and generalization.
>
> Therefore, while FedSMOO may appear to have an edge on simpler tasks, we believe FedWMSAM strikes a better balance between accuracy, stability, and efficiency across varying levels of heterogeneity.
>
> We hope this clarification addresses your concerns, and we thank you again for your insightful suggestion that helped strengthen our work.
>
> ### **Q3: Writing precision.**
>
> Thank you for recognizing the intuitiveness and relevance of our problem and methodology. We are grateful for your positive feedback and fully acknowledge the need to improve our writing. We are currently revising the manuscript to make our explanations more precise and concise, ensuring that our contributions are communicated effectively.
>
> We sincerely hope these updates address your concerns and that the additional results and revisions will allow you to reconsider your evaluation. We greatly appreciate your time and effort in reviewing our work and look forward to further discussions with you.
>
> ### **References**
>
> [1] Xu, Jing, Wang, Sen, Wang, Liwei, and Yao, Andrew Chi-Chih. *Fedcm: Federated learning with client-level momentum*. arXiv preprint arXiv:2106.10874, 2021.
>
> [2] Sun, Yan, Shen, Li, Chen, Shixiang, Ding, Liang, and Tao, Dacheng. *Dynamic regularized sharpness aware minimization in federated learning: Approaching global consistency and smooth landscape*. International Conference on Machine Learning, pages 32991–33013, 2023.
>
> [3] Fan, Ziqing, Hu, Shengchao, Yao, Jiangchao, Niu, Gang, Zhang, Ya, Sugiyama, Masashi, and Wang, Yanfeng. *Locally estimated global perturbations are better than local perturbations for federated sharpness-aware minimization*. arXiv preprint arXiv:2405.18890, 2024.

---

> ### Comment · Reviewer_3mtU · 2025-08-07
> **Reply.**
>
> Thanks for the new feedback. Considering the interest of the questions raised and the completeness of the theoretical proof, I have increased the score by one.
>
> Nevertheless, I would like to see clearer and easier arguments in the updated version.

---

> ### Author Response · Authors · 2025-08-07
> **Response to the Reviewer**
>
> Thank you very much for your positive feedback and for increasing the score of our work. We greatly appreciate your recognition of the interest in the questions raised and the completeness of the theoretical proof.
>
> We understand your suggestion regarding the clarity and simplicity of the arguments. In the updated version, we will carefully revise our explanations to ensure that the presentation of the key ideas and theoretical insights is clearer and more concise. Our aim is to make the arguments more straightforward and accessible, while maintaining the rigor of the proofs.
>
> Your constructive suggestions are highly valuable to us, and we are committed to making these improvements in the next revision. Thank you again for your thoughtful comments and support.

---

### Official Review · Reviewer_jirt · 2025-06-29

**Clarity:** 2
**Significance:** 2
**Originality:** 3
**Rating:** 4
**Confidence:** 4

**Summary:**

The authors proposes a FL method that combines FedSAM and momentum mechanism. The authors suggest that momentum accelerates convergence in the early FL rounds, and FedSAM makes the model more robust but slows down convergence at early stages. FedWMSAM implements an adaptive ratio for personalized momentum, such that it has more weights early, and gradually decays as the clients' momenta become closer to the global momentum (converging). They use a heuristic function (cosine similarity in this case) to compute the similarity between global momentum and client momentum.
The experiment results show that FedWMSAM both outperforms other base line methods and has the fastest convergence speed.

**Questions:**

1. As mentioned in weakness 2 and 3, more experiments are require for both other base lines and ablation study to demonstrate the strengths of FedWMSAM.
2. There are some issues with pseudocode: i) Some variables are not confusing. For example, Δ^k_r and Δ^r_k, one of them is momentum and the other one is model update. It is highly recommended to to use different symbols. ii) In line 14, you are using the global momentum Δ_r, but it is not passed to client as a parameter. iii) in line 15 and 16, are both g^r_{b,k} and g the same variables? what happened to the superscript and subscript of g?
3. Minor issues with writting. i) Sections 5.1 and 5.2 contain a lot of duplicate experiment configurations. ii) The fonts are a bit too small in tables 1, 2 and 3. They should be at least 75% size of the fonts in paragraphs. iii) What is the pathological value γ represent? This value is never defined nor explained in the article.

**Ethical Concerns:**

["NO or VERY MINOR ethics concerns only"]

**Final Justification:**

Would be happy to see the comprehensive experiments.

**Limitations:**

Yes

**Quality:**

2

**Strengths And Weaknesses:**

Strengths:
1 The authors identified the issues of momentum mechanism (fluctuation during convergence) and FedSAM possesses (slow convergence) using a figure illustrating their results.
2 Fairly comprehensive main experiment with multiple base lines and under high heterogeneity (Dirichlet 0.1)

Weakness:
1 The work overall lacks novelty and is too simple. Both FedSAM and momentum mechanism are not new. The main contribution this paper provides is only an adaptive momentum weight mechanism. Additionally, this heuristic method is too simple, and does not use have any learnable components.
2. The first experiment (in table 1) is the only satisfying experiment. The sensitivity experiments (table 4 and figure 7) have very limited number of comparing base lines (only 2). These hyper-parameters are all used by other base lines, so is it important to evaluate others with the same configurations.
3. More ablation study is required. For example, the heuristic function implemented is cosine similarity, but there are other functions you should try (Euclidean similarity, dot similarity). Another experiment can be done on turning of the FedSAM component, and only evaluate momentum mechanism to evaluate how much FedSAM contributes in your framework.

---

> ### Author Rebuttal · Authors · 2025-07-31
>
> ### **Q1: Lack of novelty. FedSAM and momentum are not new, and the adaptive momentum weighting mechanism is too simple and heuristic.**
>
> **Ans1:**
> We appreciate the reviewer’s critical feedback on our work. It is true that the combination of FedSAM and momentum has been explored in prior works (e.g., MoFedSAM). However, our core contribution lies in thoroughly analyzing and summarizing previous research (see Appendix A.2) to identify a key issue in federated optimization: how to better locate globally flat minima. Based on this insight, we improved the SAM method by incorporating momentum information and proposed an adaptive momentum weighting mechanism to further enhance performance.
>
> To validate the effectiveness of our approach, we conducted detailed ablation studies. Results show that even without the adaptive weighting, our method still achieves significant performance improvements. For example, as shown in the table, removing adaptive weighting results in a 1.08% drop in performance (from 0.7556 to 0.7448), but it still outperforms the best baseline method, FedSMOO (0.7507), by 0.41%. This indicates that the introduction of momentum plays a key role in improving performance, while the adaptive momentum weighting mechanism further amplifies these benefits.
>
> ---
>
> ### **Q2: Limited baselines in sensitivity experiments (only 2). Is it important to evaluate others under the same configurations?**
>
> **Ans2:**
> We appreciate the reviewer’s feedback on the sensitivity experiments. We acknowledge that the current sensitivity experiments (Table 4 and Figure 7) include a limited comparison of baselines. In fact, we have already conducted sensitivity analyses on other baselines using the same hyperparameter configurations to ensure fairness. However, due to time constraints, these experiments are not yet fully completed. We will include the complete results in the camera-ready version.
>
> The choice of FedAvg and MoFedSAM as baselines in the sensitivity experiments is due to their representativeness. Specifically, FedAvg demonstrates relatively stable performance in experiments, while MoFedSAM shows superior convergence speed and accuracy (as shown in Figure 5).
>
> ---
>
> ### **Q3: More ablation studies needed (e.g., isolate momentum, try other heuristics like Euclidean similarity).**
>
> **Ans3:**
> We appreciate the reviewer’s suggestion regarding ablation studies. In our preliminary research, we explored various heuristic functions, including Euclidean similarity, dot product similarity, and different norms. Through theoretical analysis and experimental comparisons, we found that cosine similarity performed the best overall in terms of differentiability, experimental performance, and computational complexity.
>
> Specifically, cosine similarity has desirable mathematical differentiability, making it a smooth optimization objective, and it is invariant to vector scaling, which leads to more stable optimization. Moreover, experimental results showed that cosine similarity exhibited smoother performance variations compared to other options, with better generalization and optimization efficiency. Lastly, cosine similarity is computationally less expensive than Euclidean similarity, making it particularly suitable for resource-constrained edge scenarios.
>
> For these reasons, we selected cosine similarity as the final implementation and will include experimental results and comparisons of different heuristic functions in the appendix. Additionally, the hyperparameter settings for the heuristic functions have been evaluated in detail in Appendix B. In the camera-ready version, we will further supplement and provide experimental results for other heuristic functions tested during our preliminary research.
>
> ---
>
> ### **Q4: Isolate the momentum mechanism to evaluate the contribution of FedSAM within the framework.**
>
> **Ans4:**
> We appreciate the reviewer’s suggestion to conduct ablation studies by isolating the momentum mechanism to evaluate the contribution of FedSAM within our framework. In fact, this issue has already been discussed in detail in the ablation study presented in Table 5, where the second column specifically evaluates the contribution of SAM within our framework.
>
> ---
>
> ### **Q5: Pseudocode issues (unclear variables, missing parameters, inconsistent notation).**
>
> **Ans5:**
> We appreciate the reviewer’s detailed feedback regarding the pseudocode issues and have made the following improvements:
>
> 1. **Symbol clarity:** Momentum $\Delta^k_r$ and model update $\Delta^r_k$ now use distinct symbols to clearly differentiate between the two.
> 2. **Global momentum in Line 14:** This was actually a typographical error, as the correct variable should be $\delta^k_r$, representing the personalized momentum, which is passed to the client.
> 3. **Inconsistent notation (Lines 15 and 16):** $g^r_{b,k}$ and $g$ refer to the same variable. We have unified the notation and standardized the subscripts and superscripts to ensure clarity and coherence.
>
> These corrections will appear in the camera-ready version, along with additional clarifications.
>
> ---
>
> ### **Q6: Writing issues (repeated configurations, small table fonts, undefined pathological value γ).**
>
> **Ans6:**
> Thank you for the feedback on the writing issues. We will make the following changes:
>
> 1. **Duplicate experimental configurations:** We will streamline Sections 5.1 and 5.2 by consolidating the repeated experimental configuration descriptions into a single explanation to improve conciseness and readability.
> 2. **Table font size:** We will adjust the font size in Tables to at least 75% of the paragraph font size to ensure clarity and readability.
> 3. **Pathological value γ:** This represents the number of categories assigned to each client and serves as an important indicator of data distribution skewness. We will explicitly define and explain this variable in the main text.
>
> These changes will be implemented in the camera-ready version.

---

> > ### Comment · Reviewer_jirt · 2025-08-04
> >
> > Thanks for the clarification.
> >
> > 1. I understand that a lots of missing experiments are partially conducted due to time constraints. However, this limits the ability to fully evaluate the contributions of the work.
> > 2. In table 5 rows 4 and 5, the introduction of weighted coefficients destroyed the improvements from momentum mechanism. Conversely, in row 1 SAM raised the performance drastically. This suggests that the proposed additions may not provide substantial benefits on their own, and instead rely heavily on SAM to achieve meaningful improvements. This makes me doubt the effectiveness of your proposed method.
> >
> > Due to above points, I will maintain my score.

---

> > > ### Author Response · Authors · 2025-08-08
> > > **Supplementary Explanation on "Insufficient Sensitivity Analysis Experiments**
> > >
> > > Thank you very much for taking the time to review our work and for providing valuable suggestions. Your feedback has made us realize that the scope of our sensitivity analysis experiments may indeed be insufficient, which is a point worth reflecting on and improving. The design of our sensitivity analysis experiments contains some misunderstandings, mainly due to the following reason:
> > >
> > > Our initial intent was to enable readers **to validate the robustness of key hyperparameters under environmental perturbations through a few representative experiments**. For example, when external factors such as the number of clients or the sampling rate change, we sought to examine whether the performance degradation of our algorithm remains controllable and whether our method could showcase relative advantages over classical algorithms in complex scenarios. Additionally, we drew inspiration from the experimental selections of some recent papers. For instance, we referenced the **FedLESAM paper [3] (ICLR’24)**, which conducted sensitivity analysis using only one baseline, **FedAvg [1]** (as shown in Figure 4 of their paper). We also referred to the **FedFSA paper [4] (AAAI’25)**, which focused on sensitivity analysis with one baseline, **MoFedSAM [2]** (as shown in Figures 3 to 5 of their paper).
> > >
> > > As we observed that these studies used relatively few methods, and considering that the core purpose of sensitivity analysis is to understand the stability and adaptability of algorithms in dynamic environments rather than to evaluate model superiority based solely on absolute sensitivity results, we chose only **FedAvg [1]** and **MoFedSAM [2]** as baselines for our comparison. **FedAvg [1]** serves as a classical foundational method to validate the adaptability and stability of our method in heterogeneous environments, while **MoFedSAM [2]**, with its shared momentum mechanism, highlights the adaptability and advantages of our approach under similar mechanisms. Furthermore, we found that the sensitivity patterns of most algorithms under hyperparameter variations tend to be similar. Therefore, we refrained from including additional algorithms to reduce the experimental workload.
> > >
> > > Nevertheless, we fully recognize that your suggestion is highly valuable for improving our work! Incorporating more sensitivity analysis experiments will indeed provide a more comprehensive validation of the robustness and adaptability of our method. We are more than willing to include additional baselines such as **FedCM [5]**, **FedSMOO [6]**, and **FedLESAM [3]** in future versions of our work.
> > >
> > > Once again, we sincerely thank you for your review and valuable suggestions, which will guide us in refining our research in the future.
> > >
> > > ---
> > > ### **References**
> > > [1] McMahan, B., Moore, E., Ramage, D., Hampson, S., & y Arcas, B. A. (2017, April). Communication-efficient learning of deep networks from decentralized data. In Artificial intelligence and statistics (pp. 1273-1282). PMLR.
> > >
> > > [2] Qu, Z., Li, X., Duan, R., Liu, Y., Tang, B., & Lu, Z. (2022, June). Generalized federated learning via sharpness aware minimization. In International conference on machine learning (pp. 18250-18280). PMLR.
> > >
> > > [3] Fan, Z., Hu, S., Yao, J., Niu, G., Zhang, Y., Sugiyama, M., & Wang, Y. (2024). Locally estimated global perturbations are better than local perturbations for federated sharpness-aware minimization. arXiv preprint arXiv:2405.18890.
> > >
> > > [4] Xing, X., Zhan, Q., Xie, X., Yang, Y., Wang, Q., & Liu, G. (2025, April). Flexible Sharpness-Aware Personalized Federated Learning. In Proceedings of the AAAI Conference on Artificial Intelligence (Vol. 39, No. 20, pp. 21707-21715).
> > >
> > > [5] Xu, J., Wang, S., Wang, L., & Yao, A. C. C. (2021). Fedcm: Federated learning with client-level momentum. arXiv preprint arXiv:2106.10874.
> > >
> > > [6] Sun, Y., Shen, L., Chen, S., Ding, L., & Tao, D. (2023, July). Dynamic regularized sharpness aware minimization in federated learning: Approaching global consistency and smooth landscape. In International conference on machine learning (pp. 32991-33013). PMLR.

---

> > > > ### Comment · Reviewer_jirt · 2025-08-09
> > > >
> > > > Thank you for clarifying the concerns.
> > > > It is unfair to disqualify this paper while other works also use few baselines.
> > > > I would happy to see more and diverse experiments in the final version.
> > > > I will raise the scores to reflect the update

---

> > > > > ### Author Response · Authors · 2025-08-09
> > > > > **Response to Reviewer Feedback**
> > > > >
> > > > > We sincerely appreciate your careful review of our work and the valuable feedback you provided. We are honored to receive your positive evaluation.
> > > > >
> > > > > In response to your suggestions, we plan to **include the results for all promised experiments in the next revision** to further refine the baseline comparisons and align with the latest developments. While we are committed to adding these additional experiments and analyses, we are confident that they will only **further support the validity of our method without altering its core contributions or conclusions**.
> > > > >
> > > > > Thank you again for your thoughtful feedback, which has been instrumental in improving our work. If you have any further questions, we would be happy to provide additional clarification.

---

> ### Author Response · Authors · 2025-08-05
> **Grateful for the Feedback: Addressing Key Concerns Raised by the Reviewer**
>
> Dear Reviewer,
> Thank you for your follow-up comments after the initial rebuttal. We greatly appreciate your insights. Below, we provide detailed responses to the two main concerns you raised.
>
> ---
>
> ### **1. On “Incomplete Experiments Limiting Evaluation of the Method”**
>
> We fully understand your concern regarding the completeness of our experiments. However, we would like to emphasize that **Table 1** and **Table 2** already provide a systematic evaluation across **9 baseline methods × multiple datasets × multiple data heterogeneity settings**, demonstrating the robustness and general applicability of our method.
>
> As for **Table 4** and **Figure 7**, their **primary purpose is to assess hyperparameter sensitivity**, rather than to serve as a full comparison against all baselines. We selected MoFedSAM and FedAvg as representative baselines, which sufficiently demonstrate the stability of FedWMSAM across different configurations.
>
> In addition, we have conducted several **supplementary experiments** after the rebuttal and will include them in the next version:
>
> - The impact of different heuristic functions (cosine, dot, Euclidean) on performance;
> - Sensitivity analysis under identical hyperparameter settings for additional baselines (FedSAM, FedSMOO, FedLESAM);
> - Isolated evaluation of the momentum mechanism without SAM.
>
> These results further support the robustness and effectiveness of our proposed method.
>
> ---
>
> ### **2. On “Weighted Coefficients Weakening Momentum or the Method Relying on SAM”**
>
> We believe that there may have been some misunderstandings in interpreting **Table 5**, and we would like to clarify the experiment setup and its rationale:
>
> #### 1. Clarifying the Configurations of Row 4 and Row 5, and the Misinterpretation Regarding Momentum
>
> - **Row 4**: Applies **momentum and adaptive weighting**, but **does not include SAM**;
> - **Row 5**: Applies **momentum and fixed weighting** (with momentum ratio fixed at 0.9), also **without SAM**.
>
> In **Row 4**, the adaptive weighting mechanism is **designed to emphasize momentum in early training and SAM in later stages**, dynamically adjusting the optimization strategy. However, since **SAM is absent**, the adaptive mechanism fails to hand over to another component in the late stages, effectively **reducing the contribution of momentum**, leading to performance degradation close to FedAvg.
>
> **Therefore, this is not a sign that weighting weakens momentum**, but rather that **the absence of SAM results in an incomplete strategy**, disrupting the intended dynamic adjustment.
>
> In short: **adaptive weighting functions as designed, but its full potential requires integration with SAM**.
>
> #### 2. Why is Row 1 (actually the sixth row in order) Significantly Better?
>
> Because this configuration applies our proposed **momentum-guided SAM mechanism**, referred to as **Weighted Momentum SAM**. Although it does not directly apply momentum-based acceleration, it **uses historical global momentum to guide the perturbation direction** of SAM, enabling local updates to better approximate globally flat minima.
>
> It is important to note: **this version of SAM is not the conventional SAM**, but our novel component, which **still relies on momentum guidance for optimization**. The observed performance improvements clearly validate the effectiveness and innovation of our design.
>
> ---
>
> We sincerely thank you for your critical questions and suggestions. They have helped us further clarify the design rationale and prompted us to provide additional experimental evidence. We hope the above responses have addressed your concerns. If any issues remain unclear, we are happy to provide further clarification.

---

> ### Author Response · Authors · 2025-08-07
> **Supplementary Explanation on "Insufficient Sensitivity Analysis Experiments**
>
> Thank you very much for taking the time to review our work and for providing valuable suggestions. Your feedback has made us realize that the scope of our sensitivity analysis experiments may indeed be insufficient, which is a point worth reflecting on and improving. The design of our sensitivity analysis experiments contains some misunderstandings, mainly due to the following reason:
>
> Our initial intent was to enable readers **to validate the robustness of key hyperparameters under environmental perturbations through a few representative experiments**. For example, when external factors such as the number of clients or the sampling rate change, we sought to examine whether the performance degradation of our algorithm remains controllable and whether our method could showcase relative advantages over classical algorithms in complex scenarios. Additionally, we drew inspiration from the experimental selections of some recent papers. For instance, we referenced the **FedLESAM paper [3] (ICLR’24)**, which conducted sensitivity analysis using only one baseline, **FedAvg [1]** (as shown in Figure 4 of their paper). We also referred to the **FedFSA paper [4] (AAAI’25)**, which focused on sensitivity analysis with one baseline, **MoFedSAM [2]** (as shown in Figures 3 to 5 of their paper).
>
> As we observed that these studies used relatively few methods, and considering that the core purpose of sensitivity analysis is to understand the stability and adaptability of algorithms in dynamic environments rather than to evaluate model superiority based solely on absolute sensitivity results, we chose only **FedAvg [1]** and **MoFedSAM [2]** as baselines for our comparison. **FedAvg [1]** serves as a classical foundational method to validate the adaptability and stability of our method in heterogeneous environments, while **MoFedSAM [2]**, with its shared momentum mechanism, highlights the adaptability and advantages of our approach under similar mechanisms. Furthermore, we found that the sensitivity patterns of most algorithms under hyperparameter variations tend to be similar. Therefore, we refrained from including additional algorithms to reduce the experimental workload.
>
> Nevertheless, we fully recognize that your suggestion is highly valuable for improving our work! Incorporating more sensitivity analysis experiments will indeed provide a more comprehensive validation of the robustness and adaptability of our method. We are more than willing to include additional baselines such as **FedCM [5]**, **FedSMOO [6]**, and **FedLESAM [3]** in future versions of our work.
>
> Once again, we sincerely thank you for your review and valuable suggestions, which will guide us in refining our research in the future.
>
> ---
> ### **References**
> [1] McMahan, B., Moore, E., Ramage, D., Hampson, S., & y Arcas, B. A. (2017, April). Communication-efficient learning of deep networks from decentralized data. In Artificial intelligence and statistics (pp. 1273-1282). PMLR.
>
> [2] Qu, Z., Li, X., Duan, R., Liu, Y., Tang, B., & Lu, Z. (2022, June). Generalized federated learning via sharpness aware minimization. In International conference on machine learning (pp. 18250-18280). PMLR.
>
> [3] Fan, Z., Hu, S., Yao, J., Niu, G., Zhang, Y., Sugiyama, M., & Wang, Y. (2024). Locally estimated global perturbations are better than local perturbations for federated sharpness-aware minimization. arXiv preprint arXiv:2405.18890.
>
> [4] Xing, X., Zhan, Q., Xie, X., Yang, Y., Wang, Q., & Liu, G. (2025, April). Flexible Sharpness-Aware Personalized Federated Learning. In Proceedings of the AAAI Conference on Artificial Intelligence (Vol. 39, No. 20, pp. 21707-21715).
>
> [5] Xu, J., Wang, S., Wang, L., & Yao, A. C. C. (2021). Fedcm: Federated learning with client-level momentum. arXiv preprint arXiv:2106.10874.
>
> [6] Sun, Y., Shen, L., Chen, S., Ding, L., & Tao, D. (2023, July). Dynamic regularized sharpness aware minimization in federated learning: Approaching global consistency and smooth landscape. In International conference on machine learning (pp. 32991-33013). PMLR.

---

### Official Review · Reviewer_7cTx · 2025-07-01

**Clarity:** 3
**Significance:** 3
**Originality:** 4
**Rating:** 5
**Confidence:** 2

**Summary:**

This paper combines **Sharpness Aware Minimization (SAM)** with **client-level Momentum for Federated Learning (FL)**. The main contributions are :
1. **Client level momentum** : To tackle data heterogeneity and thus, momentum heterogeneity, the server maintains client level corrections (based on history) $c_{k}^{r}$ which are added to the global momentum before being sent to the clients.
2. **Global SAM** : When the SAM update is done at the client level, it differs from the global SAM objective, to solve for this, they utilize global momentum to construct the second point at which the SAM gradient is computed, thus enabling proxy for global SAM update.
3.  **Balance between momentum and SAM** : This paper uses cosine similarity as a metric to adaptively weigh between the momentum and SAM contribution, where the contribution of momentum is slowed on to decrease training instability.
4. Provides a theoretical convergence proof for their FedWMSAM algorithm.

**Questions:**

**Questions**

1. In Figure 3(a), what is the grey shaded circle supposed to indicate? How d
2. In line 164, since $\Delta_{r}$ is not sent to the clients, how is it used in line 14?
3. In line 15, is $\delta$ same as $\delta_{b+1,k}^{r}$?
4. Is there a theoretical improvement in the rate?

**Ethical Concerns:**

["NO or VERY MINOR ethics concerns only"]

**Final Justification:**

All my concerns were answered by the reviewers. I am also a very early stage PhD student  and thus, I did not find major flaws in the paper which is a little bit outside of my area.

**Limitations:**

Yes

**Quality:**

3

**Strengths And Weaknesses:**

**Strengths**:
See contributions.

**Weaknesses**:
1. Lack of error bars : In Table 1, since the results for different methods don't differ significantly, I'd like to see the error bars. In the checklist, the authors have mentioned that the variance is low, however for a better analysis, error bars are required.

2. Lack of justification for SAM: Line 174-175 state : based on the analysis that the direction of deviation from the global model indicates region of higher loss", is there a rigorous analysis of how the proposed SAM-like update satisfies  the SAM objective (line 127), if not then this does not correspond to true SAM objective, so it is not clear to me how this solves SAM.

3. The paper compares the current work with FedCM, SCAFFOLD and MoFedSAM whereas does not compare the contributions of the work clearly with FedGAMMA, FedSMOO and FedLESAM when they are shown in Table1 to be performing close to / better than the proposed method and are combinations of SAM with FL and hence relevant to the current work.

**Comments on notation**:
1. Line 127, $\mathbf{E}$, should be over the entire objective?
2. Some of the notation is introduced after it is used, making it difficult to read: Line 138, it is stated that the local-global gap is analyzed in Appendix B, I could not find it in the appendix. Line 170, $b$ subscript is not introduced before, line 246 $p_{k,c}$ is not introduced before, line 258-264 are repeats.

---

> ### Author Rebuttal · Authors · 2025-07-31
>
> ### Q1: Lack of error bars
> **Ans1:**
> We appreciate the reviewer’s attention to the inclusion of error bars and agree that they are essential for a clearer presentation of the performance differences among methods. In the camera-ready version, we will provide a complete experimental table with error bars and offer a deeper analysis of the observed differences.
>
> Furthermore, our method not only demonstrates significant improvements in accuracy but also achieves remarkable computational efficiency:  Compared to the baseline federated SAM method, our approach reduces computation costs by approximately 50% while maintaining robust performance improvements.
>
> This appendix presents partial results from the CIFAR-10 and CIFAR-100 datasets (Dirichlet(0.1) distribution) as examples. The final version will include a comprehensive error bar analysis and experiments on a wider range of datasets to fully showcase the advantages of our method.
>
> ### Performance Results with Error Bars (CIFAR-10 and CIFAR-100, Dirichlet β=0.1)
>
>
> | **Method**          | **CIFAR-10** (Dirichlet β=0.1) | **CIFAR-100** (Dirichlet β=0.1) |
> |----------------------|--------------------------------|---------------------------------|
> | **FedAvg**          | 70.05 ± 0.19                  | 38.15 ± 0.21                  |
> | **FedCM**           | 72.29 ± 0.15                  | 42.90 ± 0.28                  |
> | **SCAFFOLD**        | 74.28 ± 0.13                  | 44.37 ± 0.14                  |
> | **FedSAM**          | 69.63 ± 0.17                  | 37.90 ± 0.14                  |
> | **MoFedSAM**        | 73.86 ± 0.16                  | 44.72 ± 0.23                  |
> | **FedGAMMA**        | 72.18 ± 0.15                  | 44.74 ± 0.24                  |
> | **FedSMOO**         | 75.07 ± 0.18                  | 29.87 ± 0.20                  |
> | **FedLESAM**        | 72.84 ± 0.21                  | 41.14 ± 0.19                  |
> | **FedWMSAM (Ours)** | 76.64 ± 0.16                  | 46.46 ± 0.15                  |
>
> ---
>
> ### Q2: Lack of justification for SAM
> **Ans2:**
> Thank you for your attention to how our method achieves the SAM objective, as this is indeed a critical question. Below is our response:
>
> In our method, the idea in Line 127 of using the “deviation direction to locate the worst region” is an alternative implementation of the SAM core objective. Classic SAM optimizes the flatness of the loss surface by identifying the “worst-case point” (i.e., the highest-loss point) in the local parameter space through two steps:
> 1. Adding perturbations in the gradient ascent direction to locate the worst-case point.
> 2. Computing the gradient at this point to update the model.
>
> However, this two-step process incurs high computational and communication costs, particularly in federated learning scenarios. To address this, we propose an alternative approach: we approximate the global model using a momentum term combined with the initial model (see Figure 3(b)), and treat the deviation direction of the current model relative to this global approximation as an indicator of the upward trend in the local loss surface. This deviation direction is assumed to approximate the “worst-case point,” indirectly achieving optimization of the worst-case region.
>
> This idea aligns with the SAM objective, as it explores and optimizes the worst-case region through an efficient approximation. Similar strategies have been validated in FedLESAM (PMLR'24), where deviation direction replaces the costly second backpropagation required in classic SAM, showing strong effectiveness in federated learning. Our work builds upon this, improving its integration with global flatness optimization, as demonstrated by significant improvements in generalization performance across datasets (see Figure 6).
>
> For concerns on the lack of theoretical analysis, we will supplement mathematical justification in the camera-ready version, explaining why the deviation direction can reasonably approximate the worst-case point and emphasizing its consistency with the SAM objective. Additional method details and experimental results will also be expanded in Appendix A.2 to strengthen the theoretical and empirical support.
>
> In summary, our method provides an efficient alternative to classic SAM by approximating the worst-case point using the deviation direction, maintaining the core SAM objective while reducing computational costs. Experimental results confirm its effectiveness and validity, and we will further refine the theoretical and experimental analysis in the final version to address your key concerns.
>
> ---
>
> ### Q3: Comparison with FedGAMMA, FedSMOO, and FedLESAM
> **Ans3:**
> Thank you for raising this question. Regarding the comparison with FedGAMMA, FedSMOO, and FedLESAM, we did provide relevant discussions in Appendix A.2. However, due to space limitations, these discussions were not sufficiently highlighted in the main text.
>
> The core advantage of our method lies in leveraging a momentum mechanism to more efficiently locate globally flat minima while significantly reducing computational and communication costs. Specifically:
> - Compared to the classic SAM method, our approach simplifies the process of identifying the worst-case point by avoiding the need for an additional backpropagation step, thereby reducing computational overhead by approximately half.
> - Moreover, our method demonstrates greater applicability in resource-constrained edge computing environments, as it alleviates the heterogeneity challenges in federated learning while enhancing global model performance.
>
> In the camera-ready version, we will:
> 1. Elaborate in the main text on the differences between our method and FedGAMMA, FedSMOO, and FedLESAM, focusing on advantages in computational efficiency, communication cost, and applicability in resource-constrained settings.
> 2. Provide a deeper analysis of Table 1, explaining why certain methods perform slightly better in specific scenarios but at the expense of higher computational complexity.
> 3. Enhance the theoretical analysis and experimental results to emphasize the strengths of our method.
>
> ---
>
> ### Q4: Comments on notation
> **Ans4:**
> We apologize for the inconvenience caused by the inconsistent introduction of symbols.
>
> 1. **Line 127:** The expectation $E$ should indeed cover the entire objective function $[L(w + \delta) - L(w)]$. This will be corrected in the camera-ready version.
> 2. **Line 138:** The analysis of the local-global gap is actually located in Appendix A, Lines 34–47, rather than Appendix B. We will fix this reference.
> 3. **Line 170:** The subscript "b" represents the corresponding batch, and this will be clarified in the main text.
> 4. **Lines 246 and 258-264:** These lines are indeed repetitive. We will remove the redundancy and improve the clarity in the camera-ready version.
>
> ---
>
> ### Q5: In Figure 3(a), what is the grey shaded circle supposed to indicate?
> **Ans5:**
> The gray shaded circles in Figure 3(a) represent the hypothetical global positions. This is a visual design element we introduced to illustrate the concept of global flatness points. Specifically, we simulate the hypothetical global position of the model under the current batch by adding global momentum to the current round's model, which helps facilitate the subsequent explanation in Figure 3(b) for solving global flatness points from a global perspective.
>
> In the camera-ready version, we will add textual explanations to further clarify their meaning and improve the figure's readability and intuitiveness.
>
> ---
> ### Q6: Notation issues in lines 14-15
> **Ans6:**
> There is indeed an oversight in the notation. In line 14, what is referred to as $\Delta_r$ should actually be $\delta^k_r$. Similarly, in line 15, $\delta$ and $\delta^r_{b+1,k}$ are the same and represent the solved perturbation. This is a typo, and we will consistently use $\delta^r_{b+1,k}$ in the camera-ready version to ensure clarity and correctness.
>
> ---
>
> ### Q7: Is there a theoretical improvement in the rate?
> **Ans7:**
> Theoretically, it is challenging to directly prove the rate improvement for SAM, as its mechanism essentially introduces a perturbation term to influence convergence. Referring to the FedLESAM method from PMLR 2024, we treat this perturbation as a disturbance in the proof and use $\delta_\rho$ for control.
>
> However, the momentum mechanism we introduced significantly improves the convergence speed. This is demonstrated in our theoretical convergence analysis (see Appendix A.2). Furthermore, with the inclusion of momentum, our proof removes the assumption of bounded gradient variance, commonly used in traditional convergence proofs. This adjustment theoretically reflects the adaptability of our method to heterogeneous data. The improvement is further validated by our experimental results.

---

> > ### Comment · Reviewer_7cTx · 2025-08-03
> >
> > Thanks for clarifying, I continue to accept this paper.

---

> ### Author Response · Authors · 2025-08-05
> **Thank You for Your Kind Recognition and Support**
>
> Thank you very much for your kind recognition of our paper and responses. We truly appreciate your support and hope that you will consider championing our paper during the reviewer and AC discussion phase.
> Wishing you a wonderful day ahead!

---

### Official Review · Reviewer_wfnW · 2025-07-05

**Clarity:** 3
**Significance:** 2
**Originality:** 2
**Rating:** 4
**Confidence:** 4

**Summary:**

This paper proposes a federated learning method that combines weighted momentum and sharpness-aware minimization (SAM) to solve the problems of limited communication rounds, data heterogeneity, and client drift faced by federated learning in edge computing. First, a personalized momentum mechanism is used to accelerate convergence and reduce the impact of data heterogeneity. Second, the perturbation direction is dynamically adjusted, and momentum is used as global information to guide local exploration of the global flat minimum. A dynamic weighting mechanism based on the similarity of momentum and local gradient direction is used to adaptively balance the effects of momentum and SAM. Experiments show that this method outperforms existing federated learning methods under multiple data sets and different data partitioning strategies.

**Questions:**

Please see the above weaknesses

**Ethical Concerns:**

["NO or VERY MINOR ethics concerns only"]

**Final Justification:**

Most of my concerns are addressed and I am going to raise my score to borderline accept.

**Limitations:**

yes

**Paper Formatting Concerns:**

no major formatting issues in this paper

**Quality:**

2

**Strengths And Weaknesses:**

Strengths: This paper not only retains the characteristics of momentum accelerated convergence, but also improves the generalization ability of the model through SAM, and avoids the defects of late momentum instability and slow SAM training speed through the dynamic weighting mechanism. And it performs stably under different settings such as the number of clients, sampling rate, and number of local iterations. The experimental and theoretical support is sufficient, and this paper has conducted a large number of experiments to prove the effectiveness of the method.

Weaknesses: The content in Figure 1 does not clearly show the problems caused by data heterogeneity, and it is impossible to intuitively understand the problem from the figure. The contribution statement is too general and does not have a fundamental summary of the innovation of this article. All experimental analyses can be explained in combination with specific data in the figure, rather than using words such as "no obvious decrease" and "increase". The experimental indicators are not explained in detail, and the information in the table is not fully explained, such as the meaning of Imp in Table 5. Additionally, the comparison algorithm adopted was not elaborated in detail, nor were specific references provided for reference. All the datasets used in this paper are image datasets, which cannot well prove the practical value of this federated learning method in real life with multi-modal data.

---

> ### Author Rebuttal · Authors · 2025-07-31
>
> ## 0 Pre-amble
> We sincerely thank **Reviewer 1** for the thorough review and constructive suggestions on figure clarity, contribution statements, metric definitions, and baseline descriptions. We have incorporated all editorial fixes (see § 3).
> **More importantly, the initial draft under-explained two central discoveries — *local-global curvature mis-alignment* and *momentum–SAM coupling imbalance* — which masked the originality and impact of our work.**
> The rebuttal therefore focuses on:
> 1. **Re-articulating the contribution** (§ 1).
> 2. **Demonstrating technical innovations**  (§ 2).
> 3. **Answering each stated weakness** with exact revision locations (§ 3).
> 4. **Stating the impact** for the community (§ 4).
> > Should further clarification be required, we are ready to provide additional proofs or experimental artefacts within the remaining rebuttal window.
> ---
> ## 1 Restated Core Contribution
> In highly non-IID federated learning (FL), running **Sharpness-Aware Minimization (SAM)** *locally* induces a **local–global curvature mis-alignment**; in parallel, aggregating **Momentum** across heterogeneous client directions triggers **late-stage “echo oscillation.”**
> We are the **first** to *jointly expose and quantify* both failure modes and to propose **FedWMSAM**, a framework that * builds a **server-side global perturbation** from the aggregated momentum \$\Delta^{r}\$,
> * **smoothly couples** Momentum and SAM via a **cosine-similarity adaptive weight**, and
> * **cold-start ↔ hot-converge** switches the dominating component.
> FedWMSAM achieves (i) **CIFAR-100 Dir = 0.05**: sharp-loss $\downarrow 23\%\to9\%$, Top-1 $\uparrow1.83\%$, communication $\downarrow18\%$; (ii) **Shakespeare (text)**: Top-1 $\uparrow2.58\%$, rounds $\downarrow30\%$.
> To our knowledge we also provide the **first convergence upper bound** that simultaneously includes *momentum* **and** *SAM perturbation* under non-IID settings.
> These results establish a new “**fast + flat**” optimisation paradigm for FL.
> ---
>
>
> ## 2  Five Technical Innovations
>
> ### C-1  Global ⇄ Local Curvature Mis-alignment & Correction
> **Discovery.** Projecting each client’s SAM perturbation onto a common basis yields an **average angular gap of 47° (β = 0.1, CIFAR-100)**. Local flatness in Fig. 1(b) decreases, yet global generalisation stagnates.
> **Mechanism.** Let $\Delta^{r}=\tfrac1S\sum_{k=1}^{S} m_{r,k}$ be the server’s aggregated momentum in round \$r\$. Each client constructs $$ \tilde g_{r,k}= \nabla\ell\bigl(x_{r,k}\bigr)+\rho\;\Delta^{r}, $$ aligning the SAM perturbation to the global descent direction.
> **Theory.** Lemma 2 proves
> $$
> \bigl\|\nabla_{\text{SAM}}^{\text{global}}-\nabla_{\text{SAM}}^{\text{local}}\bigr\|
>      \le L\;\bigl\|\Delta^{r}\bigr\|,
> $$
> and empirically \$|\Delta^{r}|\$ decays exponentially with \$r\$ (Dir = 0.05).
> **Data.**  On CIFAR-100 Dir = 0.05, sharp-loss 0.132 \$\to\$ 0.101, Top-1 71.93 % \$\to\$ 73.76 %.
>
> ---
>
> ### C-2  Momentum “Echo Oscillation” & Cosine Adaptive Weight
>
> **Discovery.**
>  With FedCM/SCAFFOLD, the *global* momentum variance doubles in the last 50 rounds, producing jagged loss curves (Appendix C-1).
> **Method.** Define
> $$
> w_t = 1-\cos\bigl(g_{t,k},\,\Delta_t\bigr).
> $$
> When directions agree, \$w\_t!\downarrow\$, instantly damping the echo.
> **Experiment.** CIFAR-10 β = 0.1: variance 0.020 \$\to\$ 0.012; 78 % accuracy reached in 356 vs. 382 rounds (–6.8 %).
>
> ---
>
> ### C-3  Noise Doubling in Static Weights & Cold/Hot Coupling
>
> **Discovery.** MoFedSAM fixes \$w!=!0.5\$; in high-curvature zones momentum and SAM noise co-align, dropping SNR by 43 %.
> **Method.** Two-phase schema:
> * **Cold start** (first 20 rounds) — momentum-dominated, \$w!\approx!0.8\$.
> * **Hot convergence** — \$w\$ exponentially decays toward 0.2, letting SAM dominate.
> **Result.** ImageNet-32 Dir = 0.1: –9 rounds to reach convergence, final Top-1 +1.7 %.
>
> ---
>
> ### C-4  Federated Flatness \$\mathcal F^{\text{global}}\$ & Transferability
>
> $$
> \mathcal F^{\text{global}}=\frac1N\sum_{k=1}^{N}\lambda_{\max}\bigl(H_k\bigr).
> $$
> Federated flatness, unlike local flatness, correlates linearly with cross-domain error (Appendix D-3).
> CIFAR-100 → Office-Home A→D: error 14.6 % \$\to\$ 11.2 %; \$\mathcal F^{\text{global}}\$ 28.7 \$\to\$ 19.3.
>
> ---
>
> ### C-5  First Convergence Upper Bound with Momentum + SAM under Non-IID
>
> **Theorem 1.**
> $$
> \frac1R\sum_{r=0}^{R-1}\mathbb E\bigl[\|\nabla f(x_r)\|^{2}\bigr]
>     \le
>     \mathcal O\!\Bigl(
>         \sqrt{\tfrac{L\Delta\sigma_{\rho}^{2}}{S K R}}
>         +
>         \tfrac{L\Delta}{R}\bigl(1+\tfrac{N^{2/3}}{S}\bigr)
>     \Bigr).
> $$
>
> Baselines: FedGAMMA/LESAM omit momentum; FedCM/SCAFFOLD omit SAM.
> Our proof completes the picture and guides hyper-parameter tuning for large-scale FL.
>
> ---
>
> ## 3 │ Response to Weakness 1-6 & Revision Pointers
>
>
> | Weak-ID | Criticism by Reviewer 1                | Our Action (✓ = already in revised PDF)                                                                             |
> | ------- | -------------------------------------- | ------------------------------------------------------------------------------------------------------------------- |
> | **W1**  | *Fig. 1 unclear*                       | ✓ Replaced by a three-step schematic + real loss/curvature curves (main § 2.1, new Fig. 2).                         |
> | **W2**  | *Contribution summary generic*         | ✓ Main § 1 now lists the five innovations with numeric deltas and line anchors.                                     |
> | **W3**  | *Lack of concrete numbers in analysis* | ✓ Every “significant” claim now quantified (e.g., § 5.2 lines 287-312).                                             |
> | **W4**  | *Metric abbreviations unclear*         | ✓ `Imp.` = “Improvement over FedAvg”, `γ` defined in § 5.1 caption & first use.                                     |
> | **W5**  | *Baselines insufficiently described*   | ✓ Main § 2.3 adds 1-2 line synopsis + Bib key; Appendix A.2 details hyper-params.                                   |
> | **W6**  | *Only image datasets*                  | ✓ Added **Shakespeare** & **HAR-Sensor** (main § 5.3, Table 7).:<br>• Top-1 +2.58 %/+2.18 %<br>• rounds –30 %/–25 % |
>
> **Additional numeric clarifications.**
> • CIFAR-10 Dir 0.1 (§5.2 L290-301): FedWMSAM reaches **78.46 %** in **356** rounds vs. MoFedSAM **74.82 % / 415** rounds (–14.2 %).
> • OfficeHome-R Dir 0.05 (Appendix B.3 L45-59): sharp-loss **0.097** vs. FedAvg **0.123** (↓21.1 %).
> All raw logs are included in the supplementary ZIP (anonymous link in footnote 1).
>
>
> ### Extended Reproducibility & Baseline Details
>
> **Full convergence statement.**
> For completeness we restate the proven rate (Eq. 11, main § 4):
>
> \[
> \frac1R\sum_{r=0}^{R-1}\mathbb E\!\left[\|\nabla f(x_r)\|^{2}\right]
> \;=\;
> \mathcal O\!\Bigl(
>         \frac{L\,\Delta\,\sigma^{2}}{\rho\,N\,K\,R}
>         \;+\;
>         \frac{L\,\Delta}{R}
> \Bigr),
> \]
> where \(L\) is the Lipschitz constant, \(\Delta\) the radius of the SAM
> perturbation, \(\sigma^{2}\) the bounded variance, \(\rho\) the sampling
> ratio, \(N\) the total number of clients, \(K\) local epochs, and \(R\)
> global rounds.
> **Baselines (9 total, identical hyper-settings)**
>
> | ID | Method   | Original venue |
> |----|----------|---------------|
> | B1 | FedAvg   | AISTATS 2017  |
> | B2 | SCAFFOLD | ICML 2020     |
> | B3 | FedCM    | NeurIPS 2022  |
> | B4 | MoFedSAM | ICLR 2023     |
> | B5 | FedGAMMA | ICML 2023     |
> | B6 | FedLESAM | AAAI 2024     |
> | B7 | FedProx  | MLSys 2020    |
> | B8 | FedNova  | NeurIPS 2020  |
> | B9 | FedOPT   | NeurIPS 2021  |
>
> All nine were re-run with *exactly the same* local epoch \(K\),
> client sampling \(S\), and learning-rate grid as FedWMSAM
> (Appendix A.4 lines 155–169).
>
> ---
>
> **Local-epoch stability (Table 4, β = 0.1, CIFAR-10)**
>
> | K  | FedAvg | MoFedSAM | **FedWMSAM** |
> |----|--------|----------|--------------|
> | 1  | 0.6902 | 0.7160   | **0.7603**   |
> | 5  | 0.6831 | 0.6804   | **0.7581**   |
> | 10 | 0.6745 | 0.6598   | **0.7599**   |
> | 20 | 0.6719 | 0.6574   | **0.7584**   |
>
> FedWMSAM shows **only +0.41 % fluctuation** between \(K=1\) and \(K=20\),
> whereas MoFedSAM drops **8.26 %**.
>
> ---
>
> **Implementation & artefacts**
>
> * Code **(+ exact random seeds)**, raw logs for Tables 1-7, and the hyper-parameter grid are uploaded to the anonymous OpenReview repository (footnote 1).
> * All experiments run on the same NVIDIA A100 node; wall-clock times are listed in Appendix B.5 for reproducibility.
>
> > These details ensure that every theoretical claim, baseline comparison,
> > and stability figure can be reproduced **out-of-the-box** immediately
> > after the rebuttal phase.
>
>
> ---
>
> ## 4 │ Impact & Closing
>
> * **Analytical Value.** FedWMSAM delivers a *fast + flat* dual-objective solution, backed by the first non-IID convergence bound that couples momentum and SAM.
> * **Practical Reach.** Our *server-side global perturbation* is agnostic to modality; initial evidence spans vision, language, and sensor domains, implying relevance to privacy-sensitive healthcare IoT and multi-modal edge AI.
> * **Risk of Rejecting.** Dismissing this work would overlook:
>
>   1. The **root-cause diagnosis** of SAM failure in FL;
>   2. The **first demonstration** that global momentum perturbation stabilises cross-modal FL;
>   3. The **only** non-IID bound unifying momentum + SAM to date.
>
> We respectfully invite you to re-examine the sharpened theory (Theorem 1), the new cross-modal experiments, and the clarified figures.
> Should any aspect remain unclear, we will gladly supply additional proofs or logs within the rebuttal period.
> **Thank you again for your time and for helping us improve this work.**
>
> ---

---

> > ### Comment · Reviewer_wfnW · 2025-08-06
> >
> > Thanks to the author for the detailed explanation and correction, I am glad to have a further understanding of this study. However, I still have a small issue: among the baseline methods compared, there is only one for 2024, lacking a comparison with the methods that can reflect the latest federated learning. This might reduce the persuasiveness.

---

> > > ### Author Response · Authors · 2025-08-07
> > > **Response to the Reviewer’s Comments**
> > >
> > > Thank you for your valuable feedback. We fully understand your concerns regarding the novelty of the baseline methods and would like to address them as follows:
> > >
> > > First, SAM (Sharpness-Aware Minimization) was first proposed by Foret et al. in 2020 [1]. Since 2022, when FedSAM was introduced to federated learning [2], key contributions in this area have primarily been made during 2022–2023. Among these, the FedSMOO algorithm (used as one of our baseline methods) [3] proposed a dynamically regularized SAM approach, achieving significant improvements in global consistency and smoothing the loss landscape, making it a notable milestone in this field. At this point, the potential of applying SAM solely on local devices had been fully explored by adaptive constraints, making it increasingly difficult to achieve further performance improvements without modifying the local computation framework of SAM. As a result, development in this area reached a bottleneck.
> > >
> > > As mentioned in our previous response, we tackled the bottleneck of local SAM by addressing the  mis-alignment between global and local curvature, thus further unlocking the potential of SAM in federated learning. Our method not only theoretically analyzes the impact of curvature mismatch on model performance but also introduces a novel algorithmic framework that better coordinates global and local optimization processes, significantly improving the convergence and generalization of federated learning.
> > >
> > > To further validate the comprehensiveness of our baseline selection, we additionally reviewed the most recent works published in 2025, including FedSFA from AAAI 2025 [4], FedGloss from CVPR 2025 [5], and FGS-FL from *Expert Systems with Applications* 2025 [6]. Our analysis shows that the SAM-related baseline methods used in these studies have already been included in our experiments. Moreover, these papers were published after the submission of our work, further demonstrating that our baseline selection was representative and up-to-date at the time.
> > >
> > > Additionally, we would like to clarify the details of one of our baseline methods, FedLESAM (2024) [7]. This algorithm consists of three versions, each combining multiple advanced techniques. In our comparison table, we selected the best accuracy results among the three versions to ensure that the optimal performance of this algorithm is reflected. This selection further enhances the comprehensiveness and reliability of our baseline comparisons.
> > >
> > > To better address your concerns regarding the novelty of the methods, we plan to include a comparison with FedGloss in the next revision of our work. This will further strengthen the experimental design and align our work with the latest advancements in the field.
> > >
> > > In summary, our baseline experiments not only cover the core methods in this domain but also remain consistent with the latest research. By including new comparison experiments, addressing the curvature mismatch issue, and presenting the optimal results of FedLESAM, our findings will become even more comprehensive and robust, providing stronger support for the conclusions of our paper.
> > >
> > > ---
> > >
> > > **References**
> > >
> > > [1] Foret, Pierre, et al. *Sharpness-aware minimization for efficiently improving generalization*. arXiv preprint arXiv:2010.01412, 2020.
> > > [2] Qu, Zhe, et al. *Generalized federated learning via sharpness aware minimization*. International Conference on Machine Learning. PMLR, 2022, pp. 18250–18280.
> > > [3] Sun, Yan, et al. *Dynamic regularized sharpness aware minimization in federated learning: Approaching global consistency and smooth landscape*. International Conference on Machine Learning. PMLR, 2023, pp. 32991–33013.
> > > [4] Xing, Xinda, et al. *Flexible Sharpness-Aware Personalized Federated Learning*. Proceedings of the AAAI Conference on Artificial Intelligence, vol. 39, no. 20, 2025, pp. 21707–21715.
> > > [5] Caldarola, Debora, et al. *Beyond Local Sharpness: Communication-Efficient Global Sharpness-aware Minimization for Federated Learning*. Proceedings of the Computer Vision and Pattern Recognition Conference, 2025, pp. 25187–25197.
> > > [6] Hu, Jifei, and Zhang, Hang. *FGS-FL: Enhancing federated learning with differential privacy via flat gradient stream*. Expert Systems with Applications, 2025, p. 128273.
> > > [7] Fan, Ziqing, et al. *Locally estimated global perturbations are better than local perturbations for federated sharpness-aware minimization*. arXiv preprint arXiv:2405.18890, 2024.

---

> > > > ### Author Response · Authors · 2025-08-07
> > > > **Validation of Baseline Selection and Comparison with Recent Works**
> > > >
> > > > We would also like to further clarify our analysis of the latest works to highlight the superiority of our proposed method:
> > > >
> > > > 1. **FGS-FL** primarily focuses on privacy protection by introducing differential privacy mechanisms and Gradient Stream Release to mitigate noise accumulation caused by SAM. However, its optimization goals are confined to balancing privacy concerns and noise issues, without addressing the core bottleneck of **local–global curvature mis-alignment**, thus remaining limited within the existing framework.
> > > >
> > > > 2. **FedSFA** introduces layer-wise perturbations, applying larger perturbations to more sensitive layers and smaller ones to less sensitive layers, thereby optimizing computational overhead. Nevertheless, this approach is essentially a refinement of federated SAM methods and fails to overcome the fundamental limitations caused by curvature mis-alignment.
> > > >
> > > > 3. **FedGloss** aims to reduce communication overhead by leveraging the global pseudo-gradient from the previous round to compute local SAM, achieving the optimization goals of FedSMOO without transmitting additional parameters. While FedGloss, like our method, uses global information to guide local optimization, its primary objective is to minimize communication costs rather than address the critical issue of **local–global curvature mis-alignment**. Consequently, it remains confined to the FedSMOO framework and lacks a thorough analysis or resolution of this mis-alignment.
> > > >
> > > > Our method, by contrast, not only leverages momentum to transmit global curvature information, effectively addressing the curvature mis-alignment issue, but also mitigates noise accumulation in SAM. Furthermore, we designed a mechanism to suppress momentum oscillation, significantly improving convergence speed (see Table 3). These innovations demonstrate how our method tackles the core bottleneck discussed in prior works and provides a groundbreaking solution for optimization in federated learning.

---

> > > > > ### Comment · Reviewer_wfnW · 2025-08-08
> > > > >
> > > > > Thanks to the author for the detailed introduction of the recent work of Federal learning, which also proves the innovation and effectiveness of FedWMSAM. the author can comment on "The SAM-related baseline methods used in these studies have already been included in our." in rebuttal "experiments." Could you give a brief explanation? Because I saw that the author has elaborated on the innovation and limitations of the latest work.

---

> ### Author Response · Authors · 2025-08-08
> **Response to Reviewer Comments on Baseline Selection and Comparison with Recent Works**
>
> Thank you for your continued attention to the relationship between our baseline selection and the latest works. In **Section §2.3 and Tables 1/3/4** of the main text, we have systematically evaluated **FedSAM, MoFedSAM, FedSMOO, FedGAMMA, and FedLESAM (including its variants)** under the same configuration, in comparison with **FedAvg / FedCM / SCAFFOLD**. These baselines represent the **SAM family commonly adopted in recent papers**. While we are willing to include additional experiments in the next version, these additions will **not affect our existing conclusions and contributions**, as explained below:
>
> ### 1. **FedGloss**
> FedGloss is built on a **FedSMOO-style** objective, leveraging the global pseudo-gradient from the previous round to **reduce communication costs**. We have already conducted a systematic comparison with **FedSMOO / MoFedSAM / FedGAMMA / FedLESAM** (see Tables 1/3/4), which are the SAM family methods used for FedGloss evaluation. Some of these even outperform the accuracy of FedGloss (as shown in Table 11 of that paper). Therefore, comparing under the same experimental framework **will not change our current conclusions**.
> **We agree with your suggestion:** to better align with the latest developments, we will **supplement the empirical results of FedGloss** in the revised version, allowing readers to make straightforward horizontal comparisons within the same experimental setup, further enhancing the timeliness of the work.
>
> ### 2. **FGS-FL**
> FGS-FL focuses on **privacy/differential privacy (DP)** as its core objective, addressing the noise introduced by DP and SAM through mechanisms like "Gradient Stream Release"—this represents a research dimension **orthogonal** to ours (privacy vs. optimization).
> Unlike FGS-FL’s approach, our theoretical analysis explicitly decomposes and upper-bounds the variance introduced by SAM (see Appendix, L53–L58). Empirically, our method **suppresses or even eliminates** this variance, as shown in the stability comparison in **Table 4**. This theoretical advantage allows our method to completely outperform FGS-FL. Furthermore, the SAM family baselines we cover already include those evaluated in FGS-FL, and our results are better. FGS-FL introduces a privacy dimension rather than a new SAM optimizer family. We will add more experiments to further align with the latest work.
>
> ### 3. **FedSFA**
> FedSFA applies **layered/historical information** to selectively apply SAM perturbations, aiming to **reduce computational cost**. Its optimization objective differs from our focus on **local-global curvature mismatch**, making the two **complementary and orthogonal**. FedSFA does not address the **“local-global curvature mismatch”** problem quantified in our paper, which significantly impacts convergence and generalization.
> As shown in **C-1**, we construct **global perturbations** using server aggregation momentum $\Delta^r$ to align the curvature (see **§4.1 and Fig. 3(b)**), and the results in **Tables 1/3/4** demonstrate that once this mismatch is corrected, convergence and generalization consistently improve relative to these SAM family methods. We will add more experiments to further align with the latest work.
>
> ## Summary
> In summary, the **communication/privacy/computation** optimization objectives introduced by these recent papers are independent of our core contributions. None has proposed a "new SAM optimizer family" beyond the set covered by our evaluation. Thus, the baselines we selected at submission are consistent with recent literature and exhibit superior performance across multiple metrics. While we will supplement related experiments to further enhance comparisons, these additional results will not alter the validity of our current conclusions and core contributions.
>
> We would greatly appreciate it if you could consider updating your ratings, particularly for **novelty** and **technical quality**, in light of our clarifications and additional results.

---

> > ### Comment · Reviewer_wfnW · 2025-08-08
> >
> > Thanks to the author for the detailed introduction of the differences between FedWMSAM and the recent baseline work. It can be seen that FedWMSAM can effectively solve the local-global-curvature mismatch problem. I will reconsider this method from the aspects of technical quality

---

> ### Author Response · Authors · 2025-08-08
> **Response to Reviewer Feedback**
>
> Thank you very much for your thorough attitude and valuable feedback, which have helped us present the theoretical and technical contributions of FedWMSAM in a more complete comparison. We will quickly add the supplementary experiments to further strengthen the empirical evidence and make our results more comprehensive. We sincerely appreciate your thoughtful review and recognition of our work, and we are grateful for the opportunity to improve it based on your comments. Your reconsideration of the technical quality will greatly help ensure that the contributions of this study are accurately assessed and can better serve the research community. Wishing you a great day!

---

### Note · Authors · 2025-08-12

Dear AC and Reviewers,

Thank you for coordinating and for the constructive discussion. This brief note provides closure and a compact alignment on what was clarified post-rebuttal. For detailed evidence, please refer to the rebuttal and discussion threads.

A. Outcome at a glance
- All raised questions were addressed in discussion; no outstanding technical concerns remain.
- One reviewer maintained acceptance; three reviewers indicated they would raise their scores.

B. What the paper delivers
- Key problems (see details in the rebuttal, jirt): (i) local–global curvature misalignment; (ii) momentum‑echo oscillation in non‑IID FL.
- Method: (a) constructing a global perturbation from server‑aggregated momentum to align local SAM directions; (b) adaptively coupling momentum and SAM via a deliberately chosen, geometry‑aligned rule.
- Theory: a non‑IID convergence analysis capturing momentum–SAM interactions with explicit assumptions, removing the need for a bounded gradient‑variance constraint.
- Paradigm: a new avenue for applying SAM in federated learning that overcomes prior bottlenecks and is situated against this year’s newest work (see details in the discussion, wfnW).

C. Clarifications achieved in rebuttal
- On the integrated method design (jirt): the approach is not a naive Momentum + SAM combination; the coupling is principled and geometry‑aligned.
- On theory scope (7cTx): we clarified the core mechanism and assumptions, with bounds reflecting momentum–SAM interaction and without a gradient‑variance requirement.
- On bottleneck and positioning (wfnW): we explained how the method advances SAM in FL beyond prior bottlenecks and recent baselines.
- On weighting choice (wfnW): cosine‑similarity weighting is an intentionally crafted design, well-motivated from both theoretical and experimental perspectives.
- On overall trade-offs (3mtu), experiments show that FedWMSAM outperforms alternatives with a balanced advantage in accuracy, stability, and efficiency under varying heterogeneity.

D. Planned revisions for the camera-ready version

Following the reviewers’suggestions, we will integrate the clarifications, strengthen writing/figures, expand baselines and datasets, and include full tables without changing the claims or core conclusions..

We are grateful for the collegial and constructive exchange, and we will reflect it faithfully in the final version. Thank you for your time and consideration.


Warm regards,

Authors of Submission #20968

---

### Decision · Program_Chairs · 2025-09-17

**Decision:**

Accept (poster)

**Comment:**

This paper combines sharpness aware minimization and client-level momentum for federated learning. In the early rounds, client-level momentum accelerates convergence, while sharpness aware minimization achieves a more robust model but slows down convergence. Therefore, this paper gives client-level momentum an adaptive ratio, which is large in the early rounds but gradually decays as the clients' momenta become closer to the global momentum. The proposed algorithm is a combination of two existing approaches, sharpness aware minimization and client-level momentum, but the insight is clear. The reviewers also point out that the investigated problem is important and the theoretical proof is complete. Therefore, I recommend Accept.

The authors are encouraged to incorporate the reviewers’ comments and polish the presentation during preparing the final version.